# Plastic Degradation by Extremophilic Bacteria

**DOI:** 10.3390/ijms22115610

**Published:** 2021-05-25

**Authors:** Nikolina Atanasova, Stoyanka Stoitsova, Tsvetelina Paunova-Krasteva, Margarita Kambourova

**Affiliations:** Stephan Angeloff Institute of Microbiology, Bulgarian Academy of Sciences, Acad. G. Bonchev str. Bl. 26, 1113 Sofia, Bulgaria; nikolina@microbio.bas.bg (N.A.); stoitsova_microbiobas@yahoo.com (S.S.); pauny@abv.bg (T.P.-K.)

**Keywords:** synthetic plastic biodegradation, extremophiles, thermophilic plastic degraders, psychrophilic degraders, halophilic degraders, biofilms

## Abstract

Intensive exploitation, poor recycling, low repeatable use, and unusual resistance of plastics to environmental and microbiological action result in accumulation of huge waste amounts in terrestrial and marine environments, causing enormous hazard for human and animal life. In the last decades, much scientific interest has been focused on plastic biodegradation. Due to the comparatively short evolutionary period of their appearance in nature, sufficiently effective enzymes for their biodegradation are not available. Plastics are designed for use in conditions typical for human activity, and their physicochemical properties roughly change at extreme environmental parameters like low temperatures, salt, or low or high pH that are typical for the life of extremophilic microorganisms and the activity of their enzymes. This review represents a first attempt to summarize the extraordinarily limited information on biodegradation of conventional synthetic plastics by thermophilic, alkaliphilic, halophilic, and psychrophilic bacteria in natural environments and laboratory conditions. Most of the available data was reported in the last several years and concerns moderate extremophiles. Two main questions are highlighted in it: which extremophilic bacteria and their enzymes are reported to be involved in the degradation of different synthetic plastics, and what could be the impact of extremophiles in future technologies for resolving of pollution problems.

## 1. General Features of Plastic Degradation

### 1.1. Plastics—Unavoidable Part of Our Daily Life. Negative Consequences from Plastic Accumulation and Slow Degradation

Plastics are man-made, high molecular weight organic polymers obtained from nonrenewable petrochemicals like fossil oil, natural gas, and coal. They are composed of hundreds to thousands of organic subunits (“monomers”) linked with strong covalent bonds. Plastics’ invasive entrance into human life during the last century has resulted in replacement of their natural counterparts in almost all industrial and domestic areas of human life. The versatility, durability, and remarkable adaptability of this group of materials improved society’s living standards, making life easier, safer, and more colorful. Their properties, such as light weight, low production cost, ease of manufacturing, bio-inertia, and resistance to environmental influence and microbial action, contribute to plastics’ extensive commercialization. According to the last report of Plastics Europe [1], everyday plastic use has shown an exponentially increasing trend for production and consumption, reaching about 350 million tons in 2019. However, a sharp growth rate drop of 8.5% was registered in 2020 due to COVID-19. The production level before the COVID-19 pandemic in the EU_27_ will not be reached again until 2022. Employed in the European plastics industry are more than 1.56 million people in 55,000 companies with over 350 billion euros of turnover. As can be seen from Figure 1, plastic producers are spread worldwide, the biggest contributors being Asia, Europe, and North America.

The first synthetic polymer, Bakelite, was produced in the beginning of 20th century; the true mass production of plastics thrived from the 1950s onwards. Over that period, their properties have been continuously improved. The most widely used polymer materials are polyethylene (PE), polypropylene (PP), polystyrene (PS), polyvinyl chloride (PVC), polyurethane (PUR), poly(ethylene terephthalate) (PET), poly(butylene terephthalate) (PBT), and nylons [2]. Currently, more than 5300 grades are produced for plastic commerce with a range of chemical additives including plasticizers, pigments, stabilizers, surfactants, and inorganic fillers [3]. Plastics have a wide range of applications in the industries for food and packaging, pharmaceuticals, agriculture, cosmetics, detergents and chemicals, etc. (Figure 2). Synthetic plastics have taken an impressive position in the packaging sector as a replacer of cellulose-based wrapping materials and now account for around 40% of the plastics produced in Europe [4]. 

Millions of tons of plastic accumulate annually as increased solid waste in terrestrial or marine environments, amounting to 20–30% (by volume) of municipal solid waste; they thus pose serious hazards for nature [5]. Today, less than 10% of all plastics are recycled [6], 24% are incinerated for energy production, and the remaining ~60% are not recovered [7]. About half of them have been estimated to accumulate in landfill sites worldwide, and others completely escape the collection system. This continual plastic pollution is a result of illegal dumping of industrial or household waste and the poor storage or transportation of such waste. Additional factors that contribute to plastics’ accumulation in the environment are their poor recycling, low repeatable use, and unusual resistance to environmental and microbiological action. It is well known that plastic pollution causes adverse effects on various ecosystems, soil fertility, the aesthetic look of cities and the environment, and on human and animal health [8]. A negative impact of plastic disposal on wildlife was ascertained for a large number of biological species including marine birds, sharks, fur seals, sea turtles, cetaceans, etc. Filling the world with discarded plastic has led to the development of social sensitivity to the impact of land pollution. As a result, the quantity of packaging waste sent to recycling has increased by 92% since 2006; however, this is still not enough, especially bearing in mind annually increasing plastic production [1].

Environmental degradation called “aging” delays plastic accumulation. It includes various mechanical and chemical mechanisms of treatment and depends on several factors. Mechanical aging depends on temperature, solar light, and moisture and causes changes in plastic bulk structure, such as cracking, discoloration, changes in shape or optical characteristics, and flaking. Chemical effects refer to changes at the molecular level by chemically oxidizing or disrupting the long polymer chain into new molecules, usually with significantly shorter chain lengths. As environmental aging is a slow process, mechanical or chemical man-made recycling is a main approach for waste treatment [9]. However, mechanical recycling releases organic and inorganic impurities in the waste, and chemical recycling is accompanied by the use of toxic and expensive chemicals [10]. Biodegradation seems to be the most effective process for waste disposal, offering specificity in attacking plastics as well as being a cheap and efficient process that does not produce secondary pollutants [11]. 

### 1.2. Microbial Degradation of Plastics

Biodegradation is the process of degradation of large polymer molecules by groups of living organisms, some of which break down the polymer chain into oligomers and monomers. Others are able to use these products, converting them to simpler waste compounds, and still others are able to use the excreted wastes. Microbial degraders and their metabolic enzymes are among the environmental agents that participate in the degradation process, which results in a conversion of the carbon in the polymer chains into smaller biomolecules or into carbon dioxide and water [12]. In this way, they contribute to soil fertility, decrease plastic accumulation in the environment, and reduce the cost of waste management. Furthermore, biodegradable plastics could be useful for the production of valuable metabolites (monomers and oligomers) [13]. The biodegradation rate is strongly influenced by several polymer properties, such as the presence of branching and additional functional groups that promote higher hydrophilicity; the relative share of crystalline and amorphous regions; the presence of polar covalent bonds, such as ester or amide bonds, instead of carbon–carbon bonds; the molecular weight and length of the carbon chain; the size and form of the substrate (powder, fibers, pellets, films); environmental agents like UV, pH, temperature, and moisture; and the enzyme characteristics [14]. Additionally, the rigidity could be increased by antioxidants and stabilizers added during plastic production. According to the rate of biodegradation, plastics are divided in two main groups: biodegradable, characterized by a higher rate process, and non-biodegradable, the biodegradation of which is slow or mediated by a still-unknown process. Most currently used fossil-based plastics are non-biodegradable, e.g., PE, PP, PS, and PVC. As large molecules of polymers cannot enter directly into the microbial cell, microorganisms responsible for primary degradation carry out extracellular degradation, and the received intermediates are further degraded by secondary degraders. According to Dussud and Ghiglione [15], biodegradation occurs in four main steps (Figure 3).
-Bio-deterioration—microbial metabolic activity provokes plastic cracks and aggravates physical properties or changes the microstructure of the matrix by pH change as a result of the released acid or biofilm formation.-Bio-fragmentation of the long polymer chain—the activity of enzymes produced by microorganisms leads to oligomer release.-Degradation of oligomers to monomers—oligomers enter inside the cells, and secondary degraders assimilate them as a carbon source, thus increasing the microbial biomass.-Assimilation of oligomers and excretion of completely oxidized metabolites to H_2_O, CO_2_, N_2_, and CH_4_. 

In spite of the hope placed on biodegradation as an innovative approach for resolving the plastic disposal problem, biodegradation is still a slow process [16]. That is why since the 1980s scientists have searched to design materials that do not affect the environment significantly and are susceptible to microbial attack. Plastic biodegradation could be enhanced by the addition of natural polymers or protein hydrolysates, optimization of the medium content and conditions for cultivation of degrading microorganisms, or gene expression for hyperproduction of the degrading enzymes [17]. Biodegradation by various microbes, such as heterotrophic bacteria and fungi, is reported for both biodegradable and non-biodegradable polymers [18]. 

### 1.3. Standard Testing Methods

Commonly used testing methods for detection of microbial activity include (Figure 4):Evaluation of visible changes in plastics such as appearance of holes or cracks, changes in surface view or color, or formation of biofilms. Scanning electron microscopy (SEM) and atomic force microscopy (AFM) represent good approaches for more sophisticated observations.Registering of changes in physical polymer properties, such as mass loss and tensile strength, and chemical properties, such as molecular weight.Measurement of utilized carbon dioxide and oxygen consumption rate.Evaluation of growth by:
-measurement of the accumulated biomass, usually in minimal media with a polymer as the sole carbon source; -formation of a clear halo around the colonies that depolymerize the polymer.Enzyme assay for detection and characterization of the depolymerization products.

#### 1.3.1. Biofilm Formation on the Plastic Surface 

Current ideas of plastic biodegradation were supported by two directions of studies: firstly, analyses of the processes that take place in natural ecosystems, and secondly, laboratory experiments designed with the aim to search for microorganisms that can be used in practice in the fight against plastic pollution. In natural environments, one decisive precondition for biodegradation is bringing the plastic and the microorganisms in close contact so that the released plastic-active exoenzymes can successfully find their way to the substrate. Biofilms are biologically relevant structures appropriate for such a purpose [19]. Biofilms are consortia of microorganisms attached to and/or embedded in an extracellular polymeric matrix. In natural systems, bacteria serve as primary colonizers that may further entrap other organisms such as fungi, diatoms, etc. [20]. The result is that the biofilm community differs significantly from the microorganisms in the free-living state in the surrounding environment [21]. When the biofilm includes organisms with plastic-degrading potential, the spatial proximity of the sessile microbial cells and the low diffusion rate of macromolecules, including extracellular enzymes, through the biofilm matrix, are factors that favor biodegradation. However, surface colonization by itself is not sufficient for plastic degradation to proceed. The type of plastic and the physicochemical conditions are both significant abiotic factors [22], while the nature of the colonizing organisms is clearly decisive. 

Among the best biodegraders for natural and synthetic polymers are several mesophilic pure bacterial species such as *Pseudomonas*, *Arthrobacter*, *Corynebacterium, Bacillus, Rhodococcus, Micrococcus*, and *Streptomyces* [21]. Under laboratory conditions, bacterial isolates cultivated to produce single species biofilms showed variable success in plastic degradation. For example, single species biofilms of *Klebsiella pneumoniae* CH001 [23] and *Rhodococcus* sp. [24] promoted the degradation of PE, and the corresponding biofilms of *Pseudomonas citronellolis* and *Bacillus flexus* showed degradation activity towards PVC [25]; similarly, *B. subtilis* ET18 and *B. cereus* ET30 each formed single species biofilm on nylon and PET, causing damage to the plastic surface [26]. Combinations of two species of bacteria have been shown to enhance plastic-biodegradation potential. Lately, attention has been drawn to the application of more complicated bacterial consortia isolated from plastic-contaminated natural habitats [20]. In spite of the apparent correlation between biofilm formation and biodegradation, the molecular mechanisms of the processes remain too vague, and there is no clear-cut link between plastic depolymerization and enzymatic activities specifically originating from biofilm microbes [27]. 

#### 1.3.2. Enzymes Participating in Plastic Degradation

One of the most used approaches for monitoring of plastic degradation efficiency is the assay of enzymes involved in the process. The resistance of plastics to microbial attack results from the short time of their presence in nature not being enough for the evolution of new enzyme structures capable of effective polymer degradation [28]. The described enzymes belong to two main classes of enzymes, namely hydrolases and oxidases, and could be esterases, proteases, cutinases, dehydrogenases, or laccases (Table 1). Esterases and lipases (EC 3.1.1.X) hydrolyze plastics by ester bond cleavage in the carbon chain and are active mainly on aliphatic polyesters. High efficiency in PET degradation was demonstrated by the aromatic polyesterase synthesized by *Ideonella sakaiensis* 201-F6 [2]. Examples of bacterial esterases able to degrade polyurethanes have been reported [29,30,31]. Cutinase (EC 3.1.1.74) is a hydrolase for a variety of polymers, although initially characterized as able to hydrolyze ester bonds in the plant polymer cutin. Great potential for hydrolysis of PET has been suggested for cutinases and their homologues [10]. Proteases (EC 3.4.X) are active on peptide bonds in polyamides, such as different type of nylons [32]. 

Gram positive (*Rhodococcus* strains) as well as Gram negative (*Sphingomonas* and *Pseudomonas* strains) bacteria that produce oxydases were reported to degrade PVA or PE (Table 1). 

Currently, information for enzymes acting on high-molecular-weight plastics such as PS, PVC, PP, polyamide, and PUR is very scarce, and a single degrader is not known [27]. Reports of their degradation mainly concern polymer derivatives and/or degradation by microbial communities without enough information to identify enzymes or pathways that are responsible. Although some degradation of PVC and PP by mixed-species microbial communities was suggested based on weight loss, the observed effect could be attributed not to the polymer degradation of the main chain but to the metabolism of the relevant chemical adjunct molecules [40]. Similarly, polystyrene sulfonate, a derivative of PS, was depolymerized in the presence of certain redox mediators through the action of brown-rot fungi [41].

It is known that extremophilic microorganisms are competent producers of a range of potentially relevant hydrolytic enzymes [42]. The search for new plastic-active enzymes and microorganisms has resulted in interest in relevant extremophilic enzymes due to the changes that occur in polymer properties at extreme values of temperature, pH, salinity, pressure, reduced water content and nutrients, and high radiation.

## 2. Extreme Environments and Extremophiles

Most currently known extremophiles are either Eubacteria or Archaea. Extremophilic eukaryotes are also known; however, the boundaries of their extremophilicity are usually significantly lower, and consequently, they have less potential than extremophilic prokaryotes to influence the properties of plastics. Although members of Archaea live at the harshest conditions and a presence of archaeal representatives was identified in plastic-degrading consortia from marine samples by molecular techniques [21] a laboratory cultivation of effective archaeal degraders has not yet been reported. For these reasons, extremophilic eukaryotes and Archaea will not be objects of discussion in this review.

“Extreme” is a relative term referring to the ability of some organisms not only to endure but to actively grow in conditions that would be lethal to or too harsh for human existence. It is generally accepted that culturable microorganisms represent a very tiny part (no more than 1%) of the species present in natural ecosystems; this fraction is especially low in samples taken from extreme niches due to difficulties in reproducing such environmental conditions in the laboratory. Consequently, microbes from extreme environments represent an untapped reservoir of microorganisms, enzymes, and biomolecules for different industrial applications. Of special interest for biotechnology, biomedicine, and industrial processes are the enzymes produced by extremophiles (extremozymes) due to their activity in the extreme conditions at which their producers can grow. The unusual properties of extremozymes and metabolic features of their producers may hold the potential for resolving one of society’s biggest problems, namely plastic pollution, and suggest a new approach for bioremediation of polluted extreme environments or the development of novel processes for composting.

Thermophiles, literally heat lovers, are microorganisms that need high temperatures (between 45 and 122 °C) for their growth [43]; as such, they are organisms that grow at temperatures above those that sustain most life forms. According to their growth temperature range, thermophiles are classified in several groups. The optimal temperature for growth of hyperthermophiles is above 80 °C [44]. Typical bacterial hyperthermophiles are representatives of the genera *Aquifex* and *Hydrogenobacter* (phylum *Aquificae*) and *Thermotoga* (phylum *Thermotogae*). Extreme thermophiles grow optimally at temperatures between 65 and 80 °C and are often representatives of the genera *Thermus* (phylum *Deinococcus–Thermus*) and *Rhodothermus* (phylum *Bacteroidetes*), which were isolated from such environments. Obligate thermophiles grow between 50 and 70 °C, with an optimum of 55–65 °C, and the most abundant obligate thermophiles are representatives of the family *Bacillaceae*, such as the genera *Anoxybacillus, Brevibacillus*, and *Geobacillus* [45]. Facultative thermophiles thrive at temperatures 41–50 °C, while thermotolerant microorganisms are mesophilic microorganisms that can tolerate temperatures higher than 41 °C but grow optimally at lower temperatures. The most abundant representatives of these two groups are also members of *Bacillaceae*. Thermophiles and hyperthermophiles habituate various thermophilic ecosystems such as deep-sea black and white smokers, terrestrial hot springs and geysers, volcanoes, fumaroles, and man-made environments such as compost facilities, steam power plants, and greenhouses. Thermophiles have developed unique mechanisms for active growth in these environment niches, such as amino acid changes in the primary structure of their proteins, shorter protein length, and the participation of heat shock proteins in protein folding; additionally, they have evolved more stable membranes by incorporating branched chain fatty acids and polyamides as well as active systems for repairing DNA damage [45].

Alkaliphiles can grow in alkaline habitats (pH > 9). They are divided into two groups: facultative alkaliphiles (optimal growth at pH 7.0–9.5) and obligate alkaliphiles (optimal growth between pH 10.0 and 12.0) [45]. Mainly alkaliphiles belong to the genera *Bacillus*, *Micrococcus*, *Pseudomonas*, and *Streptomyces* [45,46]. They have developed systems to regulate the influx of protons and solutes inside the cell by changing ion distribution (e.g., Na^+^). Alkaline environments include alkaline hyper-saline lakes and some man-made environments that result from agricultural activity. 

Psychrophilic microorganisms can grow in the temperature range from −20 to 20 °C, with an optimal growth temperature below 15 °C [47]. They have adapted to low-temperature growth by a number of different strategies, including an increased amount of unsaturated fatty acids and short chain fatty acids in membranes, which prevents a loss of membrane fluidity; high synthesis of cold-shock proteins; synthesis of anti-freeze proteins that bind to ice crystals; accumulation of compatible solutes as cryo-protectants to prevent cell damage; and adaptation of psychrophilic enzymes to activity at low temperature by modifications of their primary structure. Cold environments comprise fresh and marine waters, including deep sea water, polar and high alpine soils, and glaciers, which represent more than 70% of the surface of our planet. The diversity of psychrophilic bacteria comprises genera as *Pseudomonas*, *Psychrobacter, Pseudoalteromonas*, *Colwellia*, *Arthrobacter*, etc. [48]. 

Halophilic microorganisms require salt for growth, and according to NaCl concentration in the medium for optimal growth, they are categorized as slight halophiles (1–3%), moderate halophiles (3–15%), or extreme halophiles (above 15%) [49]. The most abundant moderate and extreme halophiles are members of two genera, *Halomonas* and *Chromohalobacter* (family *Halomonadaceae*) [49]. Their natural niches are salterns, saline lakes, oceans, and coastal areas. The intracellular systems of halophiles have been adapted to avoid water losses by two unique strategies: either maintaining more water in the cytoplasm than outside of the cell by osmotic pressure balance achieved by organic compatible solutes, or maintaining high intracellular salt concentration by active salt transportation with participation of bacteriorhodopsin and ATP synthase [45]. 

As a general principle, evolving in harsh conditions has made extremozymes more rigid and resistant to proteolysis and denaturing agents such as organic solvents and detergents. An attractive feature of thermophilic and halophilic enzymes is the slowing down of the process for enzyme “aging” that allows their storage at room temperature with a longer half-life of commercial preparations. Their long life is beneficial because it potentially prevents a significant loss of enzymatic activity in slow processes such as plastic degradation. Furthermore, many plastic-contaminated niches are characterized by extreme environmental conditions such as low or elevated temperatures, high salt concentrations, acidic or alkaline pH, or high pressure. The available information for plastic degradation by extremophiles predominantly concerns moderate extremophiles. Possible reasons for this could be the short time for evolving metabolic mechanisms of extremophilic adaptation toward these difficult substrates, allied to enhanced sensitivity to substrate or product inhibition in the case of thermophiles, and the low growth rate of psychrophiles that becomes even slower in the presence of such difficult for assimilation substrates.

## 3. Plastic-Degrading Thermophilic Bacteria

The use of thermophiles for plastic degradation in the biological treatment of polluted thermal habitats is potentially advantageous because of improved substrate bioavailability and solubility as a result of the changes in physical and optical polymer properties at elevated temperatures [14]. Additional advantages of thermophilic biodegradation processes are the higher rates of enzyme activity as a result of the decreased polymer strength, the enhanced diffusion rates of organic compounds, the decreased viscosity of culture liquids, and the reduced risk of microbial contamination. Several thermophiles have shown high potential for polymer degradation due to their ability to grow and produce numerous enzymes in unusual conditions. Examples of thermophilic plastic degraders are shown in Table 2.

Polycaprolactone (PCL), a biodegradable synthetic aliphatic polyester of ε-caprolactone, is characterized by a low melting point (60 °C). Its hydrophobicity and crystallinity resemble those of conventional plastics and determine its large application instead of non-degradable plastics [13]. An effective synergy between two thermophilic bacteria isolated from compost with predominated PCL wastes at 50 °C was reported [54]. PCL degradation by a pure culture of one of the microorganisms, the thermophilic actinomycete *Streptomyces thermonitrificans*, when analyzed by PDS–GPC (gel permeation chromatography) revealed that a peak corresponding to the molecular weight of the initial PLC polymer began to decrease gradually 72 h after cultivation accompanied by the appearance of oligomeric peaks. Further degradation of the resultant oligomers was demonstrated by the lowering of the relevant peaks, suggesting that this strain could achieve extensive mineralization of PCL, resulting in a 35% decomposition of the plastic after 6 days of composting. However, when *Bacillus licheniformis* HA1, the synergistic partner isolated from the same compost was added, a significant increase in PCL degradation was observed, reaching a value of 70% after 48 h. It was suggested that while *B. llcheniformis* HA 1 alone was not able to utilize plastic, it was able to grow by degradation of the intermediates released by *S. thermodenitrificans* and the consequent altered pH. The synergistic effect resulting from the simultaneous cultivation of both thermophilic strains accelerated PCL degradation and significantly increased the portion of the decomposed polymer. It was suggested that the constant concentration of the primary degrading microorganism was a result of the low rate of plastic degradation. Complete degradation of PCL within 6 h at 45 °C by a thermophilic *Streptomyces thermoviolaceus* subsp. *thermoviolaceus* 76T-2 was reported [55]. Two PCL-degrading extracellular enzymes with molecular weights of 25 kDa and 55 kDa were secreted by this microorganism. Thermophilic actinomycetes active on polyhydroxybutyrate (PHB), PCL, or polyethersulfone (PES) were isolated from composting [56]. In a similar report, a reduction of the gravimetric and molecular weights of branched low-density (ld) PE (11 and 30% correspondingly) by a thermophilic bacterium, *Brevibaccillus borstelensis* strain 707, was registered after 30 days at 50 °C in spite of the fact that it was a poor biofilm former [50]. Although less well characterized, a thermophilic isolate, *Bacillus* sp. BCBT21, changed the properties and appearance of both high- and low-density polyethylene plastic bags at 55 °C within one month [51]. 

Polyethylene terephthalate (PET) is largely used for the production of synthetic fibers for the textile industry, and its accumulation leads to environmental pollution [7]. It becomes available for enzymatic hydrolysis at temperature about 65–75 °C due to the enhanced mobility of the amorphous sectors of the polymer chains [57]. Consequently, effective enzymatic degradation of PET could be achieved by thermostable PET hydrolases [58]. Most bacterial isolates able to degrade PET belong to the Gram-positive phylum *Actinobacteria*, mainly the genera *Thermobifida* and *Thermomonospora*, and specifically the species *Thermobifida alba* [59], *Thermobifida halotolerans* [60], and *Thermomonospora curvata* [61]. Approximately 50% degradation of low-crystalline (lc) PET was achieved at 55 °C after 3 weeks of action by the extracellular polyester hydrolase TfH secreted by the thermophilic bacterium *Thermobifida fusca* [28]. When the recombinant *T. fusca* cutinase TfCut2 was over-expressed in *B. subtilis*, it was the dominant protein in the supernatant, which after 42 h of cultivation at 37 °C was able to degrade lcPET film [62]. The amorphous regions were almost completely degraded (97.0%) within 120 h at 70 °C at a linear rate of 20–22% per day. Both endo- and exo-type scissions of the PET polymer chains were confirmed by NMR analysis. Two types of cutinases that share 93% identity in amino acid sequence were isolated from *T. fusca* [63]. They metabolized the synthetic polyesters and possessed high thermostability. Despite the high similarity between the two enzymes, only Tfu0883 was able to degrade PET at 60 °C, suggesting that the amino acid sequence differences are located at the substrate binding site. This enzyme demonstrated a good tolerance to surfactants, a superior stability in organic solvents, and a superior thermostability. A cutinase able to degrade PET and PCL was cloned from a fosmid library of a leaf–branch compost metagenome and expressed in *E. coli* [64]. It hydrolyzed various fatty acid monoesters optimally at pH 8.5 and 50 °C. Its half-life was 40 min at 70 °C and 7 min at 80 °C. The anaerobic thermophile *Clostridium thermocellum* was applied as a whole-cell biocatalyzer combining the enzyme production and hydrolysis of PET in a single step [52]. The weight loss of an amorphous PET film was over 60% after a 14-day incubation at 60 °C. The observed degradation rate of >2.2 mg/day was higher than that for the described whole-cell mesophile *Ideonella sakaiensis* (>1.4 mg/day) [2].

Nylon is the generic name for a related group of synthetic polyamides characterized by a high resistance to degradation due to crystalline morphology received as a result of strong intermolecular hydrogen bonds between the polymeric chains [65]. Its numerical nomenclature depends on the number of carbon atoms in the monomers used for their manufacture. The thermophilic bacteria *Anoxybacillus rupiensis* Ir3 used nylon 6 in a minimal medium as a sole carbon and nitrogen source at 65 °C [53]. *Geobacillus pallidus* strain 26 degraded nylon 12 and 6 at 60 °C, but was not able to degrade the more crystalline nylon 66 [32].

All of the above reported thermophilic plastic degraders belong to the group of facultative and obligate thermophiles, with thermophilic *Bacillaceae* being a good source of enzymes for plastic transformation bioprocesses, and to the best of our knowledge equivalently competent extreme and hyperthermophiles are currently not known. Similarly to mesophiles, thermophilic *Bacillaceae* members are a good source of enzymes for plastic transformation bioprocesses.

## 4. Alkaliphilic Degraders 

pH is another environmental factor that can affect both the solubility and softening of plastics. However, the ability to degrade synthetic polymers by acidophilic bacteria has been scarcely investigated. At the same time, acidic pH shortens the life of some plastic products used in bleaching processes at low pH. Information concerning the degrading capability of alkaliphiles is scarce. Low-density (ldPE) polyethylene (PE) was degraded by bacterial strains isolated from hyperalkaline water samples (pH 11) from a spring in the Philippines [66]. Nine strains were isolated after enrichment in a synthetic medium supplemented with ldPE as a sole source of carbon and pH-adjusted to 11. They were phylogenetically affiliated with *Bacillus krulwichiae*, *B. pseudofirmus*, *Prolinoborus fasciculus*, and an unclassified *Bacillus* sp. Pure cultures of the isolates reduced the polymer weight by up to 9.9%, 8.3%, 5.1%, and 6.3% respectively after 90 days of growth without any pre-treatment of ldPE. In each case, a slow and constantly proliferating biofilm was observed. Furthermore, a significant increase in the effectiveness of PE degradation by a bacterial community isolated from the same spring was observed in the presence of iron oxide nanoparticles (IONPs) [67]. The effect of IONPs was attributed to the properties of the nanoparticles, such as magnetism and electrostatic charge, altering bacterial motion through signal transduction. As a result, higher hydrophobicity of the consortium with IONPs and higher adhesion to the plastic surface were demonstrated. The addition of the IONPs facilitated biofilm formation by the participating strains at pH 11. The weight of the residual polymer was reduced by 18.3% and 13.7% in the presence and absence of IONPs, respectively, after 60 days of incubation. Two strains identified as *Bacillus pseudofirmus* and *B. agaradhaerens* were isolated from the biofilm, both of which were classified as obligate. The effectiveness of PE degradation by the pure strains was investigated. Corresponding polymer weight losses of 6.46% and 8.36% were observed for pure cultures of the isolates in the absence of IONPs, which increased to 9.62% and 11.32% in the presence of IONPs. The same albeit enhanced trend was observed with both the unsupplemented and IONP-supplemented biofilm communities, the former of which even in the absence of the iron oxide nanoparticles was more effective than either of the strains isolated from the community. 

Currently available information suggests that most identified alkaliphilic plastic degraders are obligate alkaliphilic species of the family *Bacilliaceae*, a trend shared with the known thermophilic plastic-degrading eubacteria. 

## 5. Halophilic Degraders 

There is growing evidence for the bioremediation of plastics in marine and other natural saline environments, such as salt marshes, as well as in salt-rich industrial wastewaters. Most of the characterized halophilic microorganisms have been found to be moderate or only slight extremophiles, with species of the genus *Erythrobacter* being predominant. Some examples of halophilic marine microorganisms able to degrade plastics are shown in Table 3. Significantly, the role of multi-species microbial biofilms in promoting plastic degradation in such environments has become increasingly recognized in recent years. A survey of different niches of seawater in the Western Mediterranean Sea [68] consistently found that in each sampled area, not only the highest number but also the highest density of bacteria was detected attached to plastic debris when compared with both sessile bacteria attached to other organic particles and free-living bacteria. The plastic debris typically consisted of polyethylene (PE) (72.2%), followed by polypropylene (PP) (18.0%) and polystyrene (PS) (2.8%), as revealed by FTIR analysis. By characterizing the operational taxonomic units (OTUs) in the total DNA extracted from each of the samples, it was found that the dominant microorganisms freely living in water belonged to *Alphaproteobacteria* (45.0%, mainly *Pelagibacter* sp.), followed by *Cyanobacteria* (24.3%, mainly *Synechococcus* sp.), *Flavobacteria*, and *Gammaproteobacteria* (11.3% and 11.1%, respectively). In contrast, equivalent analyses confirmed that the dominant microorganisms sourced from plastic debris were *Cyanobacteria* (40.8%, mainly *Pleurocapsa* sp.) and *Alphaproteobacteria* (32.2%, mainly *Roseobacter* sp. and *Erythrobacter* sp.), while the dominant microorganisms sourced from other organic particles were *Alphaproteobacteria* (25.9%, mainly *Erythrobacter* sp.), *Gammaproteobacteria* (25.0%, mainly *Alteromonas* sp.), and *Cyanobacteria* (17.9%, mainly *Synechococcus* sp.). It was suggested that the relatively large recorded presence of *Cyanobacteria* species on plastic debris was not only determined by their established important role in biofilm formation [69], but also possibly by some activity towards plastic debris. Specifically identified strains belonged predominantly to two genera, *Calothrix* sp. and *Pleurocapsa*, which are known halophiles often isolated from marine environments. Analysis of the OTUs sourced from plastic debris also confirmed that *Erythrobacter* species were predominant (43%) amongst the detected hydrocarbonoclastic bacteria, and that two other moderate halophilic genera, *Hyphomonas* and *Phorimidium*, were present in significantly higher levels than in samples sourced from water and organic particles. 

Comparison of the biological diversity in biofilms formed on PS samples incubated in Black Sea water at 10 °C and industrial water from a petrochemical plant revealed different community composition [70]. It was suggested that the significant difference in salinity may be reflected in the active growth of slight halophiles in sea water samples (1.86% salinity) and non-extremophiles in industrial water samples (~0.1% salinity). High-throughput sequencing of the V3–V4 region of the 16S rRNA genes were used to characterize the microbial composition of the biofilms. *Erythrobacter (Alphaproteobacteria)* increased during the incubation and became a dominant genus in the biofilm growing on seawater-incubated PS samples after 60 days of incubation, while the portion of other genera like *Pelagicoccus (Verrucomicrobiota), Pseudohongiella (Gammaproteobacteria)*, and *Planctomicrobium (Planctomycetota)* decreased. The cyclic formation and removal of biofilms throughout the 60-day incubation period resulted in a more intensive biodegradation of the polymer. The participation of a putative enzyme phenylacetaldehyde dehydrogenase (EC 1.2.1.39) was suggested in PS degradation on the basis of iVikodak metagenomic analysis [71]. Based on the metabolic pathways of different taxa, this enzyme was most likely to be affiliated with the detected species of the genera *Pseudomonas*, *Arenimonas*, and *Acidovorax* in the industrial water samples and with detected species of the genera *Erythrobacter*, *Maribacter*, and *Mycobacterium* in the seawater samples. The same enzyme is known to be involved in phenylalanine metabolism [72], but its relationship to the mechanism of PS degradation is poorly understood and requires further clarification.

Polyethylene terephthalate biodegradation was investigated in Black Sea, fresh, and industrial waters with respective salinities of 18.6, 0.09–0.3, and 1.3 g/L [73]. Investigation of microbial diversity in consortia isolated from these environments revealed a universal presence of representatives of the phyla *Bacteroidetes*, *Gammaproteobacteria*, and *Alphaproteobacteria*, albeit in different proportions. PET degradation in samples from the industrial water was attributed to the specific presence of the genera *Pseudomonas* and *Acidovorax* due to the detected presence of genes responsible for terephthalic acid degradation in their genomes. This hypothesis was confirmed by photomicrography that revealed a local change in PET surface after exposure in industrial water, while similar changes were not observed for marine PET samples.

The involvement of sessile microbial communities in the biodegradation of other types of plastic has been recognized. Biofilm formation on PCL and PVA was observed by SEM (Figure 5) for a microbial community from Pomorie salterns (28% salinity), Southeast Bulgaria [74].
ijms-22-05610-t003_Table 3Table 3Various literature reports on plastic biodegradation by psychrophilic and halophilic microorganisms.Plastic Degradation Type PolymerMicroorganism Isolation SourcePhysico-Chemical Parameters of Environment Effectiveness of DegradationReferenceNon-biodegradablePolyethylene (72.2%) + PP (18.0%) and PS (2.8%)*Cyanobacteria* (*Calothrix*, *Pleurocapsa*, *Phormidium*), *Erythrobacter*Western Mediterranean Sea 3.87% salinitynot reported[68]PolystyreneCommunity, *Erythrobacter*Black Sea water1.86% salinitynot reported[70] Polyurethane CommunityBaltic Sea10 °C, pH 8.0, 1.86% salinity19% weight loss for PU-A, 4% weight loss for PU-B after 12 months [75] BiodegradablePolycaprolactone*Shewanella*, *Moritella*, *Psychrobacter*, and *Pseudomonas*Kurile and Japan Trenchesdepth of 5000–7000 m, 4 °Cnot reported[76] 

## 6. Psychrophilic Degraders

Approximately 70% of the Earth is covered by marine water, 90% of which maintains a constant temperature of ~5 °C regardless of latitude [77]. As a result, a predominant part of the Earth’s surface provides a suitable environment for the development of psychrophiles. Often marine microorganisms are multiple extremophiles, such as psychro-halophiles, psychro-piezophiles, or psychro-alkaliphiles [77]. Most of the plastic-degrading bacteria identified in cold environments belong to the genera *Shewanella*, *Moritella*, and *Psychrobacter*, class *Gammaproteobacteria*.

Some examples of psychrophilic marine degraders are shown in Table 3. While it has been proposed that such microorganisms play a role in degrading the plastics in marine ecosystems, there is currently insufficient knowledge to understand the influence of plastics on microbial life, function, and community structure in these environments [21]. However, it is recognized that attachment to surfaces and growth within the resultant biofilms constitute an important survival adaptation of bacteria under the conditions of cold marine environments. Biofilms formed on plastic debris provide a more secure environment protected from environmental hazards, and the biofilm structure promotes the switch-on of metabolic reactions in the microbial assemblages that may result in fragmentation and even breakdown of the debris [78]. The existing tendency of additive supplementation to the most explored synthetic plastics for better biodegradability determined the interest in the biodegradation of such substituents. The Baltic Sea is a cold niche, reaching the freezing point in winter and still very cool in summer, and is thus favorable for a growth of psychrophilic bacteria. Consequently, it was chosen as the environment to compare the biodegradability of uncross linked poly(ethylene-butylene-adipate) (PU-A) and slightly crosslinked poly-(ε-caprolactone) (PU-B), two poly(ester-urethane) plastics with different structural characteristics [75]. Samples of both plastics were exposed to the seawater in Gdansk harbor (about 10 °C and pH ~8). After 12 months, a loss of tensile strength, discoloration, and cracking were recorded with both materials, but were more significant for un-crosslinked PU-A. The efficiency of degradation for PU-A was 19% weight loss, while for PU-B it was 4% weight loss. An active enzymatic hydrolysis of ester bonds was suggested as a primary step in the degradation of un-crosslinked molecules of PU-A, followed by crystallinity and network structure attack. 

According to the United Nations Environment Program (UNEP) [79], natural plastic degradation in surface waters is usually performed at a higher rate than in bottom waters due to reduced sunlight (UV) penetration and colder temperatures in depth waters. At the same time, a decrease in plastic strength due to increased pressure at depth water should be considered. Thirteen strains that degrade PCL were isolated from deep seawaters at depth of 300–600 m [76]. The isolates were incubated at 4 °C and 50 MPa. Among the isolates, eight belonged to *Moritella*, three to *Shewanella*, one to *Psychrobacter*, and one to *Pseudomonas*. Investigation of the effects of temperature and hydrostatic pressure on the growth of the isolates revealed that all of the *Shewanella* and *Moritella* isolates are polyextremophilic psychrophilic and high-pressure adapted bacteria. Their degradation activity was confirmed by the halo-formation method and transmission electron microscopy.

Some examples of marine microorganisms (halophiles and/or psychrophiles) able to degrade plastics are shown in Table 3.

Thirteen PCL degrading bacteria belonging to the genera *Shewanella, Moritella*, *Psychrobacter*, and *Pseudomonas* were isolated from the Kurile and Japan Trenches at a depth of 5000–7000 m [76]. Three of the isolates demonstrated typical piezophilic growth and four were piezo-tolerant. Based on their activity on PCL and growth profiles under different hydrostatic pressures, an active microbial role in the degradation of aliphatic polyesters under deep-sea bottom conditions was suggested. Deep-sea sediment bacterial isolates identified as *Pseudomonas* sp. and *Lysinibacillus* sp. showed biofilm-related potential for high density PE biodegradation even within a short, 24 h test interval [80]. These microorganisms produced increased extracellular matrixes that improve cell adhesion to plastic surface. A good correlation between biofilm biomass and changes to PE structure was observed. 

## 7. Conclusions

Recently, scientific and technological interests have been focused on developing highly effective enzyme processes for managing the negative impact of plastic pollution on human and wild life. The use of extremophilic microorganisms and their enzymes is a promising way to address this very serious social concern. Extreme conditions contribute to plastic degradation by extremophiles by a higher enzyme rate as a result of plastics’ softening and disruption of plastics’ mechanical integrity. Biodegradation by extremophiles under unique environmental conditions or in waste treatment facilities opens the way for reducing disposed plastic waste. Despite the relatively short evolution time, a significant number of extremophilic microorganisms have adapted to grow in these environments by plastic degradation and in such a way play an important role in the biological remediation of contaminated extreme environments. An intensive search for the identification of new microorganisms from extreme niches is quite promising because of their ability to develop different adaptation mechanisms. However, their industrial application is still limited because of the technical difficulties in their cultivation, their lower biomass and productivity yield, the reduced specific activity of their enzymes (the rigidity of the thermophilic enzyme molecule limits the enzyme–substrate complex rate formation), and the sensitivity of enzyme synthesis and activity to substrate or product inhibition. The available information reveals that extremophilic degraders belong mainly to slight and modest extremophiles. Two reasons could be suggested for this: firstly, very extreme conditions significantly decrease biological diversity and correspondingly the chance for evolving of microorganisms able to degrade plastic; and secondly, the growth rate at these very extreme conditions is usually low, and bacterial growth cannot be totally supported by such hard-to-degrade polymers. It is possible that after a longer evolutional adaptation to plastics’ availability in nature, extreme degraders will also appear. Another objective could be the development of relevant metagenome technologies that permit searching for and expression of genes for novel enzymes or variants of known enzymes with improved relevant properties directly from environmental metagenomes, as well as modifying the enzymes by a genetic engineering approach. The prospect for developing effective processes on the basis of extremophilic bacteria and their properties was the objective of the current review, which to the best of our knowledge is a first attempt at summarizing the restricted information on plastic degradation by extremophiles.

## Figures and Tables

**Figure 1 ijms-22-05610-f001:**
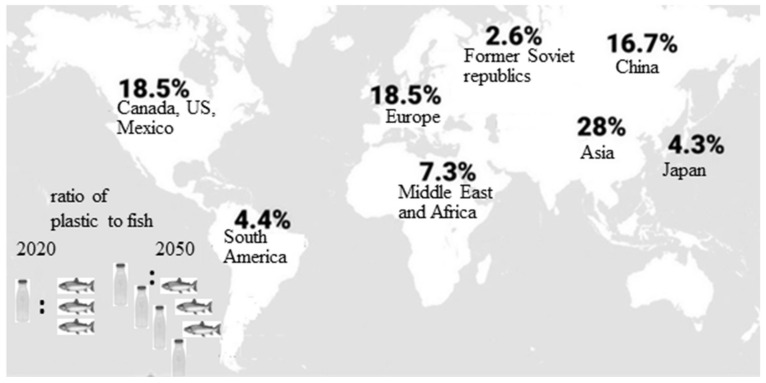
Worldwide distribution of plastic producers. Data according to [1].

**Figure 2 ijms-22-05610-f002:**
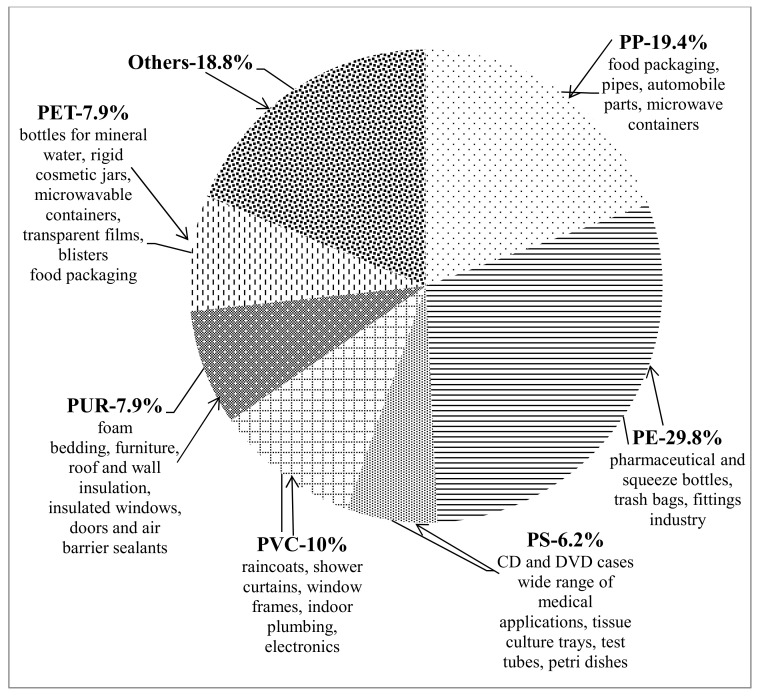
Demand distribution and use of different plastics (data according to [1]).

**Figure 3 ijms-22-05610-f003:**
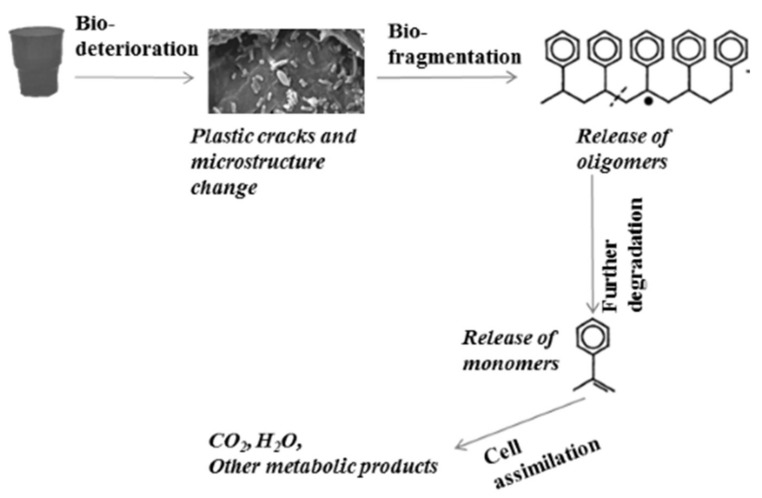
Overview of plastic degradation.

**Figure 4 ijms-22-05610-f004:**
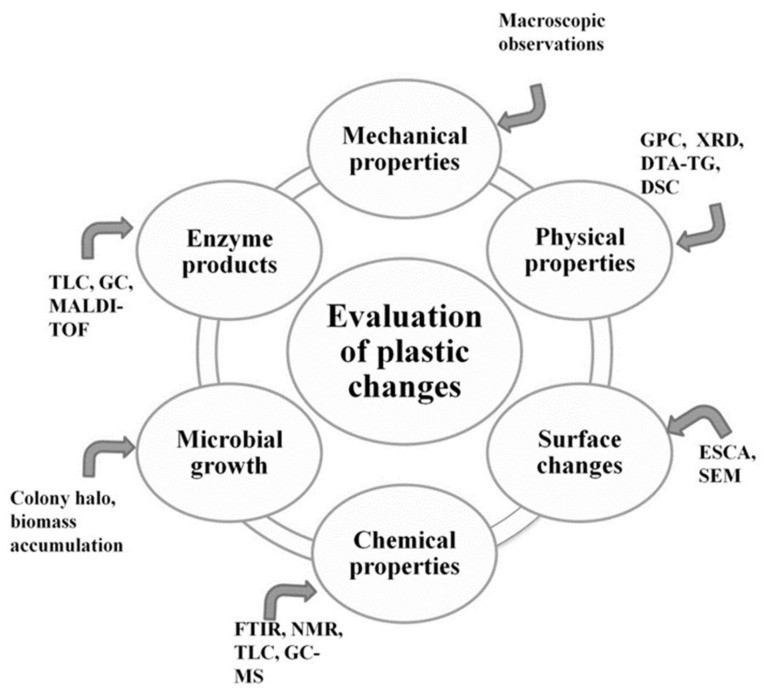
Analytical techniques for monitoring the extent and nature of plastic degradation. Abbreviations: SEM-scanning electron microscopy; GPC-gel permeation chromatography; XRD-X-ray diffraction; DTA TG-differential thermal analysis and thermogravimetric analysis; DSC-differential scanning calorimetry; ESCA-electron spectroscopy for chemical analysis; FTIR-Fourier transform infrared spectroscopy; NMR TLC combined application of nuclear magnetic resonance and thin-layer chromatography; GC-MS-gas chromatography/mass spectrometry; MALDI-TOF-matrix-assisted laser desorption/ionization time of flight.

**Figure 5 ijms-22-05610-f005:**
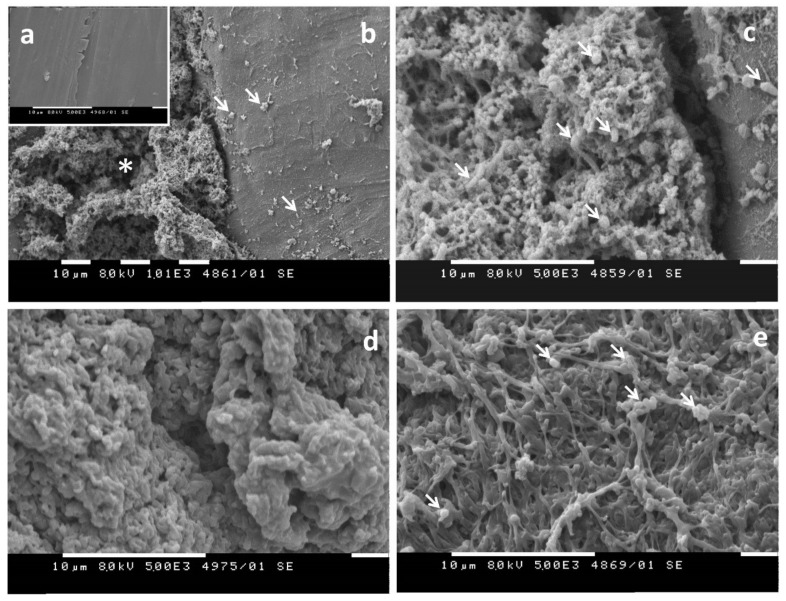
SEM images showing the interaction between bacteria from a Pomorie Salterns (PS) community and plastics after two weeks of co-incubation [74]. (**a**) Surface relief of a PCL sample incubated in the absence of bacteria; (**b**,**c**) Polycaprolactone (PCL) co-incubated with PS community. At lower magnification (**b**), a deep groove in the plastic (asterisk) is visible filled with filamentous material formed as a result of partial degradation of the plastic. Adherent bacteria are present on the comparatively unaltered part of the surface (arrows). (**c**) higher magnification shows biofilm bacteria inside the groove on the plastic (arrows). (**d**) Surface relief of a polyvinyl alcohol (PVA) sample incubated in the absence of bacteria; (**e**) PVA co-incubated with PS community. The structure of the polymer is loosened, and bacteria adherent to polymer filaments are observed (arrows).

**Table 1 ijms-22-05610-t001:** Some examples of plastic-degrading bacterial enzymes.

Enzyme Class	Enzyme Group	Enzyme Source	Type of Biodegraded Plastic	Reference
Hydrolases that split ester bonds	Esterase	*Streptomyces* sp. SM14	PET	[33]
	Esterase	*Bacillus subtilis*	PU	[29]
	Esterase	*Alicycliphilus* sp.	PU	[30]
	Aromatic polyesterase	*Ideonella sakaiensis* 201-F6	PET	[2]
	Esterase E3576	Commercially available by Proteus	PU	[31]
	Lipase	*Alcaligenes faecalis*	PCL	[34]
Hydrolases that act on carbon–nitrogen bonds	amidase E4143	Commercially available by Proteus	PU	[31]
	6-aminohexanoate-cyclic-dimer hydrolase, 6-aminohexanoate -dimer hydrolase and endo-type6-aminohexanoate-oligomer hydrolase	*Flavobacterium* sp. KI72	6-aminohexano-ate, an intermediate product of nylon	[35]
Oxydase	PVA dehydrogenase	*Sphingomonas* sp. strain 113P3	PVA	[36]
	Alkane hydroxylase	*Pseudomonas* sp. E4	PE	[37]
	Laccases	*Rhodococcus ruber*	PE	[38]
	Monoxygenases	*Rhodococcus* sp. TMP2	PE	[39]

**Table 2 ijms-22-05610-t002:** Various literature reports on plastic biodegradation by thermophiles.

Plastic Degradation Type	Polymer	Microorganism	Isolation Source	Temperature for Polymer Degradation	Effectiveness of Degradation	Reference
Non-biodegradable	Polyethylene	*Brevibaccillus borstelensis* strain 707	Soil	50 °C	11% after 30 days	[50]
		*Bacillus* sp. BCBT21	Composting agricultural residual	55 °C	44% decrease of average MW of the polymer for 30 d	[51]
	Polyethylene terephthalate	*Thermobifida fusca*		55 °C	≈50% decrease of the average MW of polymer for 3 weeks	[28]
		*Clostridium thermocellum*		60 °C	60% after 14 days	[52]
	Nylon	*Anoxybacillus rupiensis*	Hydrocarbon contaminated soil	65 °C	Optical density ≈1.8 after 7 days growth on nylon 6	[53]
		*Geobacillus thermocatenulatus*	Soil	60 °C	Decrease in nylon 12 and nylon 66 MW from ≈ 40,000 to ≈15,000 over 20 d	[32]
Biodegradable	Polycaprolactone	Consortium—*Streptomyces thermonitrificans* PDS-1 + *Bacillus licheniformis* HA1	Compost	50 °C	70% (compost as a substrate) for 48 h	[54]
		*Streptomyces thermoviolaceus* subsp. *thermoviolaceus*	Soil	45 °C	100% (0.1% substrate) for 6 h	[55]

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
