# Peer review of "Plastic Degradation by Extremophilic Bacteria"

_ijms, 2021, doi:10.3390/ijms22115610_

Round 1

Reviewer 1 Report

Comments included in the attached file. 

Author Response

All cited references need to be checked carefully, both in text and in the Bibliography. This  is for 2 reasons: 

  1. In the manuscript as submitted there is a discrepancy between the citation

.          number in the text and the corresponding number in the Bibliography. The reason      

.          for this appears to be: 

                           Text.                              Bibliography 

                             [1]……corresponds to…….[1] 
                                                                 one of these references in the Bibliography 
                                                                 Is not included in the text, and as a consequence                               [9]……corresponds to……[10]   

all in this sequence are 

therefore -1 compared   

to the Bibliography 

                             [25]…..corresponds to……[26] 
                                                                        [27] this reference (Pinto et al) is not included                                                                                 in the text, and as a consequence 
                             [26]…..corresponds to……[28] 

all in this sequence are 

therefore -2 compared   

to the Bibliography 

                             [62]…..corresponds to……[64] 

  1. The revised text has suggested the introduction of new additional references

.          which will consequently require a revision of the numbering of the other                       

.           subsequent references both in text and in the Bibliography.         

Sorry for the discrepancy in the numbering of references. This version was carefully checked and hope now numbering is correct.

Page 1 Line 36:     human life.  Versatility …. 
Replace with:        human activity.  Versatility …… 

The word life was replaced by activity”

Page 1 Line 39:     production cost, easy manufacturing ……  Replace with:        production cost, ease of manufacturing ….. 

“Ëasy”was changed by “ease of”

Page 1 Line 45:     rate in 2020:  -8.5%.  Replace with:        rate in 2020 of -8.5% 

“of” was added

Page 2 Line 5:      1, plastic producers are spread worldwide with a biggest deal of  Asia 

Replace with:       1, plastic producers are spread worldwide, the biggest contributors                             

.                                being Asia 

The phrase was changed according to the suggested way

Page 2 Line 16:     based wrapping materials. Their packaging application accounts for            

.                             around 40% of the plastic 

Replace with:         based wrapping materials , and now account for around 40% of the.         

.                              plastics produced. 

The phrase was changed according to the suggested way

Page 5 Line 13:      a slow process [15]. That’s why ….. 

Replace with:          a slow process [15]. That is why ….. 

That’s” was replaced by “That is”

Page5 Lines 23                                                                                                                        

+ 24:                      The resistance of plastics against microbial attack is a result of a       

.                              short evolutionary time of their presence in nature that is not                    

.                              enough for design of  

Replace with:         The resistance of plastics to microbial attack results from the                    

.                              short time of their presence in nature not being enough for the                 

.                              evolution of  

Replaced

Page 5  Line 26:     Enzymes involved in plastic degradation processes belong to the               

.                              groups 

Replace with:         Characterised enzymes known to be involved in plastic degradation          

.                              processes comprise a relatively small group 

This part was changed according to the advice of another reviewer.

Page 5 Lines29                                                                                                                         

+ 30:                      ………………………… active mainly on aliphatic polyesters. Bacterial 
                              polyurethane esterases have been reported to degrade polyurethanes 
Replace with:        …………………………..active mainly on aliphatic polyesters. Examples 

.                             of bacterial esterases able to degrade polyurethanes have been                

.                              reported [19]. 

Replaced

Page 6  Lines 1                                                                                                                                           

+ 2:                       Cutinase (EC 3.1.1.74) is known as a hydrolase for a variety of                   

.                             polymers although initially has been described as able to …. 

Replace with:       Cutinase (EC 3.1.1.74) is a known hydrolase for a variety of                           

.                            polymers although initially characterised as able to …. 

The sentence was changed.

Page 6 Line 6:       An new additional reference is needed for any examples of proteases                 .                             able to hydrolyse polyamide-type synthetic polymers such as nylons -               .                             otherwise it needs to be made clear that while proteases can hydrolyse 

.                             natural polyamides (such as proteins), this ability is yet to be confirmed   

.                             with synthetic polyamides (as is implied by the text included in Page 6    

.                             lines 8-10 incl). 

The reference Tomita et al., 2003 was included

Page 6 Line 8:       Still the information ….. 
Replace with:        Currently, the information …… 

Replaced

Page 6 Line 10   

+ 11:                                                  ……….[20]. The reports for their degradation                 

.                                 concerns mainly  polymer …………..                                                  

Replace with:                                 ……….[20]. The reports of their degradation                  

.                                 concern mainly  polymer …………..   

Replaced

Page 6 Line13:             responsible for their accumulation. Although a degradation of ..   Replace with:               responsible. Although some degradation of …………  

Replaced

Page 6 Lines16 

+ 17:                             degradation but to the attack to the chemical additives in their           

.                                 molecules [17]. As an example polystyrene sultanate …      

Replace with:            degradation , but to the metabolism of the relevant chemical                

.                                 adjunct molecules [17]. Similarly, polystyrene sultanate …      

Replaced

Page 6 Lines 20  

+ Line 21:                    Seeking for new plastic active enzymes and microorganisms    .        

.                                determines the interest toward extremophiic enzymes due to the           

.                                 change………….. 

Replace with:              The search for new plastic active enzymes and microorganisms    .        

.                                  has resulted in interest in relevant extremophiic enzymes due to the          

.                                  changes that occur   

Replaced

Page 6 Line 23              

+ 24                            …… water content and nutrients, high radiation. Such enzymes are      

.                                  produced by extremophilic microorganisms 

Replace with:             ……. content of water and other nutrients, and high radiation. It is          

.                                  known that extremophilic microorganisms are competent producers .                           

of a range of potentially relevant hydrolytic enzymes [ * ] 

                                   * a new relevant reference is required here  eg ‘Extremophiles for         

.                                    hydrolytic enzyme production: biodiversity and potential                     

.                                    biotechnological applications’  D. Kour et al.,                                        

.                                    doi.org/10.1002/9781119434431.ch16 

The sentence “Such enzymes…” was removed.

Page 6 Line 26:                                  ….. methods for registration of microbial 
Replace with:                                      ….. methods for establishing relevant microbial 

This phrase was changed

Page 6 Line 28:                       ……. changes in plastic like appearance of holes or 
Replace with:                         ……. changes in plastic such as appearance of holes or

Replaced

Page 6 Line 31:                ………………a good approach for sophisticated 
Replace with:                    ………………a good approach for more sophisticated 

Replaced

Page 6 Line 42:            5.    Registration of products …………. 

Replace with:               5.     Detection and characterisation of products ……….. 

Replaced

Page 7 Figure 4:         hallo  Replace with:               halo 

Page 7 Lines 8  

+ 10:                           ….………….…….the substrate. The biologically relevant structure                           

.                                  appropriate for such a purpose is the biofilm [23]. Biofilms represent       

.                                  consortia of microorganisms attached to surfaces and embedded in    

.Replace with:             ….………….…….the substrate. Biofilms are biologically relevant           

.                                   structures appropriate for such a purpose[ 23]. Biofilms are                  

.                                  consortia of microorganisms attached to and/or embedded in an    .

Replaced                  

Page 7 Lines17                              ………rate of macromolecules, including                         

+ 18:                          enzymes, through the biofilm matrix, are prerequisites for                                              

Replace with::                 ………rate of macromolecules, including extracellular                      

.                                enzymes, through the biofilm matrix, are all factors that favour

                        Replaced

Page 8 Lines 1                                                                                                                            

+ 2:                                                      ……. conditions are significant factors                            

.                                  [26], the nature of the colonising organisms is decisive. 

Replace with:                                    ……. conditions are both significant abiotic factors    
                                   [26], while the nature of the colonising organisms is clearly decisive. 

Replaced

Page 8 Lines 8                                                                                                                        

+ 14:                                   ….. degradation. For example, Klebsiella pneumonia CH001          

.                        [27] and Rhodococcus sp. [24] formed biofilm which was related with             

.                degradation of PE, biofilms by Pseudomonas citronellolis and Bacillus flexus          

.                showed degradation activity towards PVC [28], B. subtilis ET18 and B.             . 

       cereus ET30 formed biofilms on nylon and PET causing damage of the                   

.      plastic surface [29]. Combinations of two species of bacteria were                      

.    shown to enhance the plastic-biodegradation potential. Lately, the attention has 

Replace with: .            ….. degradation. For example, single species biofilms of               

.                          Klebsiella pneumonia CH001 [27] and Rhodococcus sp. [24] promoted   

.                          the degradation of PE, and corresponding biofilms of Pseudomonas         

.                           citronellolis and Bacillus flexus showed degradation activity towards       

.                           PVC [ * ] ; similarly, B. subtilis ET18 and B. cereus ET30 each formed   

.                           single species biofilms on nylon and PET causing damage to the                  

.                plastic surface [29]. Combinations of two species of bacteria were shown    .     

.               to enhance the plastic-biodegradation potential [ ** ]. Lately, the attention has                          

 Replaced

.                [ * ]. Reference [28] currently shown in text actually refers to original                 

.                          research reported in the review currently listed as [29] in the                   

.                          Bibliography. The correct reference to the relevant original scientific        

.                          paper is:  Giacomucci L. et al.,  New Biotechnol. 52, 35 - 41 (2019)n.    

                              [ ** ] There is currently no relevant reference cited in text, but this is            
.                             required.  A suitable paper for this purpose would be Pinto, M.  et al       
.                             PLoS ONE 2019, 14. - currently listed as [27] in the Bibliography. 

Relevant bibliography is listed

Page 8 Lines 17 

+ 18:                                                     ………… formation and biodegradation, to                  

.                                     the  present moment the molecular mechanism ….  

Replace with:                                    ………… formation and biodegradation,                     

.                                      currently the molecular mechanism ….   

Changed

Page 8 Line 20:                   …….and  enzymatic activities originating from biofilm 

Replace with:                     …….and  enzymatic activities specifically originating from 
biofilm 

Replaced

Page 8 Line 23:                   Mostly extremophiles belong to Bacteria and Archaea. 

Replace with:          Most currently known extremophiles are either Eubacteria or Archaea. 

Replaced

Page 8.Lines 25  extremophilicity is usually significantly lower and consequently could not 

+ 26:                     influence the plastic  properties to such an extent as prokaryotes do. 

Replace with:  extremophilicity is usually significantly lower and consequently they            

.                       have less potential than extremophilic prokaryotes to influence the               

.                       properties of plastics.  

Replaced

Page 8 Line 29:                                  …..archaeal degraders is not yet reported. 

Replace with:                                     …..archaeal degraders has not yet been reported. 

Replaced

Page 8 Line 30:         That’s why extremophilic eucaryotes ….. 

Replace with:            For these reasons extremoophilic eucaryotes …..

Replaced

Page 8 Lines 33        

+ 34:                                                 …..to grow in lethal or too harsh for human 

                               existence conditions.  It is generally ……….. 

Replace with:                              …..to grow in conditions that would be lethal or too         

.                              harsh for human existence. It is generally ……….. 

Replaced

Page 8 Lines 36 

- 38:                      species available in natural ecosystems and in the extreme niches this     

.                             amount is especially low due to difficulties to reproduce specific              

..                            environmental conditions in the laboratory. Microbes from extreme 

Replace with:      species present in natural ecosystems : this fraction is especially low.   

.                           in samples taken extreme niches due to difficulties in reproducing           

.                           such environmental conditions in the laboratory Consequently,     

       .                    microbes from extreme 

  Replaced

Page 8 Lines 40                                         ……..industrial applications. Among 

- 42:                 biomolecules the extremophilic enzymes (extremozymes) are of             .                           

special interest for biotechnology, biomedicine and industrial                    .                           

processes due to                 

Replace with:                                               ….. industrial applications.  Of special             

.                         interest for biotechnology, biomedicine and industrial processes are           

.                         the enzymes (extremozymes) produced by extremophiles due to 

Replaced

Page 8 Line 45:      producers represent a challenge for resolving ………. 
Replace with:          producers may hold the potential for resolving ………. 

Replaced

Page 8 Lines 49 

- 52:                   Thermophiles are heat loving microorganisms that need high 

                        temperatures for their growth between 45 and 122oC [30]. Thermophiles        

.                      literally heat lovers are organisms that grow at temperatures above those   

.                       that sustain most life forms. According to their growth temperature 

Rep[lace with:  Thermophiles, literally heat lovers, are microorganisms that need high 
                        temperatures (between 45 and 122oC) for their growth  [30] : as such,     

.                       they are organisms that grow at temperatures above those that sustain    

.                       most life forms. According to their growth temperature range,   

Replaced

Page 9 Line 1:       thermophiles are classified in several groups,   facultative thermophiles  Replace with:        they are classified in several groups ; facultative thermophiles 

This paragraph was rewritten.

Page 9 Line 5:      80oC. Growth of hyperthermophiles …. 
Replace with:       80oC  , while growth of hyperthermphiles ……. 

This paragraph was rewritten.

Page 9 Line 7:     ecosystems such as deep-sea and terrestrial hot springs, geysers, 
Replace with:      ecosystems such as deep-sea black smokers, terrestrial hot springs and    

.                           geysers,  

Replaced

Page 9. Lines 12    

- 14:                     proteins, shorter protein length, participation of heat shock proteins in          

.                         protein folding; stabilisation of membranes by branched chain fatty acids             

.                          and polyamides; active system for repairing of DNA damages [32]. 

Replace with:     proteins, shorter protein length, and the participation of heat shock               

.                         proteins in protein folding; additionally they have evolved more stable          

.                         membranes by incorporating branched chain fatty acids  and                         

.                         polyamides ,  and active systems for repairing DNA damage [32]. 

Replaced

Page 9 Line 15:                                                           …..They are divided in 

Replace with:                                                              …..They are divided into

 Replaced

Page 9 Line 18:      They have developed systems for influx of ……. 

Replace with:          They have developed systems to regulate the influx of ……. 

Replaced

Page 9 Lines 20 

+ 21:                                                                          …… alkaline soda lakes,                                                

Replace with:               man-made environments as a result of agricultural activity.                                 

                                                                               … alkaline soda lakes, and some  

                                      man-made environments that result from agricultural activity. 

Replaced

Page 9 Line 24:                                                                      …..growth by an increased … 
Replace with:       …..growth by a number of different strategies including an increased…

     Replaced

Page 9 Line 26:             ……… membrane fluidity ;  high synthesis …..    Replace with:                ……… membrane fluidity ,  high synthesis …..  

Replaced

Page 9 Lines 27  

- 29:                       proteins ; synthesis of anti-freeze proteins that bind to ice crystals ;       .                           

accumulation of compatible solutes as cry-protectants to prevent cell         .                        

damage ; adaptation of psychrophlic enzymes activity to low temperature 

Replace with:   proteins , synthesis of anti-freeze proteins that bind to ice crystals ,          

.                          accumulation of compatible solutes as cry-protectants to prevent cell   .                          

damage , and adaptation of psychrophlic enzymes to activity at low

Replaced

Page 9 Line 39:     strategies ,  maintaining  more water ……… 
Replace with:        strategies; either maintaining  more water ……… 

Replaced

Page 9 Line 44:          Evolving in harsh conditions ….. 

Replace with:             As a general principle, evolving in harsh conditions ….. 

Replaced

Page 9 Line 46:         solvents and detergents. Attractive feature of ……. 
Replace with:            solvents and detergents. An attractive feature of ……. 

Replaced

Page 9 Line 47:     enzymes is slowing down of ……… 
Replace with:         enzymes is the slowing down of ……… 

Replaced

Page 9 Line 49:     ….. Their long life prevents a ….. 

Replace with:         ….. Their long life is beneficial because it potentially prevents a ….. 

Replaced

Page 9 Line 50:                ……………..like plastic degradation. Many plastic- 

Replace with:                   ……………..like plastic degradation. Furthermore, many plastic- 

Replaced

Page 9 Line 53:                      ……or high pressure. Analysis of …… 

Replace with:                          ……or high pressure. However, analysis of …… 

The whole sentence was changed

Page 10. Lines 2 

7:                                                             …… moderate extremophiles . The reason for                            

.                          very limited information for degradation by extremophiles living in very                 

.                               extreme conditions could be short time for evolving metabolic          .                           

mechanisms of extremophilic ………………..difficult substrates ;         .                           

the enhanced sensitivity. ……..………………………… in the case of         .                           

thermophiles, low growth rate ……….           

Replace with:             …… moderate extremophiles ,whereas there is  . The reason for                            .                         very limited information for degradation by extremophiles living in very                  
.               extreme conditions. Possible reasons for this discrepancy, which are relevant     

.          to all categories of extremophiles, could be the short time for evolving metabolic 

Replaced

               mechanisms of extremophilic ……………………difficult substrates , allied to               

.                enhanced sensitivity   ……..…………………………………….  in the case of          

.                 thermophiles, and the low growth rate ………. 

Corrected

Page 10 Line11:     ………………….thermal habitats suggests an improved 

Replace with:       …..thermal habitats is potentially advantageous because of improved 

Replaced

Page 10 Line15:           are the higher enzyme turnover rate as a ……..  Replace with:                are the higher rates of enzyme activity as a …… 

Replaced

Page 11. Table 1: This Table needs to be reformatted. Where it is not possible to include     

.                             whole words, they need to be hyphenated in a ‘sensible’ way, eg 
                              Degrad-        Poly-            Biodegrad-     Polycapro- 
                               ation              ethylene      able                 lactone 

                              In addition, Mw needs to be amended to MW

Changed 

Page 12 Line 7:     degradation by a thermophilic actinomycete ……. 

Replace with:        degradation by a pure culture of one of the microorganisms, the                

.                             thermophilic actinomycete …….. 

Replaced

Page 12 Line 8:      analysed by PDS-GPC ………….  Replace with:         when analysed by PDS-GPC ……. 

Replaced

Page 12 Line 9:                                …………molecular weight of the used PLC began to 
22                 decrease gradually at 72 h of its cultivation accompanied by appearance 
                         of oligomeric peaks. Further degradation of oligomers was demonstrated 
                         by lowering of their peaks suggesting that this strain could achieve full 

                         mineralization of PCL and decomposed 38% of plastic after 6 days of 
                         composting. However, when Bacillus licheniformis HA1 isolated from the 
                        same compost was added a significant increase in PCL degradation was 
                        observed reaching a value of 70% after 48 h. B. licheniformis HA 1 alone 
                        was not able to utilise plastic, however it grew by degradation of the  
                        intermediates and adjusted pH. Synergistic effect in the simultaneous 
                        cultivation of both thermophilic strains accelerated PCL degradation and               

.                       increased significantly the portion of the decomposed polymer. The 
                        constant concentration of the primary degrading microorganism was 
                        suggested to be a result of the low rate of plastic degradation as a rule. 

Replace with:            …………molecular weight of the initial PLC polymer began to              

.                         decrease gradually 72 h after cultivation commenced,  accompanied          

by the appearance of oligomeric peaks. Further degradation of the resultant oligomers was  demonstrated by the lowering of the relevant peaks suggesting that this strain could   
achieve extensive mineralization of PCL , resulting in a 35% decomposition of the plastic 
after 6 days of composting. However, when Bacillus licheniformis HA1 , the synergistic   
partner isolated from the same compost was added  , a significant increase in PCL 
degradation was observed reaching a value of 70% after    48 h. It was suggested that 
while B. llcheniformis HA 1 alone was not able to utilise plastic, it was able to grow by 
degradation of the intermediates released by S. thermodenitrificans and the consequent 
altered pH. The synergistic effect resulting from the simultaneous cultivation of both 
thermophilic strains accelerated PCL degradation and increased significantly the portion of  the decomposed polymer. It was suggested that the constant concentration of the primary  degrading microorganism was a result of the low rate of plastic degradation.

Replaced

Page12 Line 25:    Two PCL-degrading enzymes …….. 

Replace with:        Two PCL-degrading extracellular enzymes ……..

Replaced

Page 12 Line28:                    ……[42]. Reduction of gravimetric and 

Replace with:                      ……[42]. In a similar report a reduction of the gravimetric and 

Replaced

Page 12 Line 32:           was a poor biofilm-former [35].  A thermophilic isolate ……… 

Replace with:           was a poor biofilm-former [35]. Although less well characterised, a            

.                                 thermophilic isolate ……

Replaced

Page 12 Line 43:   Thermomonospora, species Thermobifida alba [45], ….. 

Replace with:         Thermomonospora, specifically the species Thermobifida alba [45],  …

Replaced

Page 12 Line 45                                         ………………after 3 weeks action of the                       

- 49:                   polyester hydrolyse TfH synthesised from a thermophilic bacterium               

.             Thermobifida fascia [18]. The recombinant T. fusca cutinase TfCut2 was over                

.              expressed in B.subtilis as the dominant protein in the supernatant                            

.             after 42 h of cultivation at 37oC which was able to degrade lcPET film   

Replace with:                                            ……………after 3 weeks of action by the              

.           extracellular polyester hydrolyse TfH secreted by the thermophilic bacterium              

.         Thermobifida fascia [ * ] When the recombinant T. fusca cutinase TfCut2 was over   

.          expressed in B.subtilis, it was the dominant protein in the supernatant   , which                           

.             after 42 h of cultivation at 37oC was able to degrade lcPET film  

v Replaced

               [ * ]  Correct reference citation required.  -  unlikely to be [18] 

Corrected

Page 13 Line13:    Nylon is a synthetic polyamide characterised …….. 

Replace with:      Nylon is the generic name for a related group of synthetic polyamides        

.                           characterised …….. 

Replaced

Page13 Line15:           … intermolecular hydrogen bonds between molecular chains …….  Replace with:        … intermolecular hydrogen bonds between the polymeric chains …….

Replaced

Page 13 Line 16   numerical nomenclature depends from the ratio between the number of 

- 18:                      carbon atoms in the diamine and dibasic acid monomers used for their         

.                             manufacturing. 

Replace with;       numerical nomenclature depends on the number of                                  

.                             carbon atoms in the monomers used for their                                              

.                             manufacture. 

Replaced

Page 13 Lines 20 

+ 21:                                                                     ………………. nylon 12 and 6 at 60oC,                               

.      however was not able to degrade nylon 66, more crystalline than other two nylons. 

Replace with:                                                   ………………. nylon 12 and 6 at 60oC,                                

.                but was not able to degrade the more crystalline polymer nylon 66 [39]. 

Replaced

Page13 Line 23:    The reported thermophilic ……. 

Replace with:         All the above reported thermophilic ………. 

Replaced

Page13 Lines 24                ……obligate thermophiles and to the best of our knowledge                    

- 27:                              extreme and hyperthermophiles are not known to the best of our 

                          knowledge. Similarly to mesophiles thermophilic Bacillaceae members         

.                         are a good source of enzymes for plastic transformation bioprocesses. 

Replace with:               ……obligate thermophiles, with thermophilic Bacillaceae being a  

.    good source of enzymes for plastic transformation bioprocesses . and to the best of

our knowledge equivalent competent extreme and hyperthermophiles are currently     .    not known. 

Replaced

Page 13 Lines 29                 pH is another environmental factor the affects solubility and                    

- 31:                   softening of plastics. To the best of our knowledge reports for a capability   

.                          to degrade synthetic polymers by acidophiles (excepting fungi) are not      

.                          available and alkaliphiles are scarcely investigated. …..    

Replace with:      pH is another environmental factor that can affect both the solubility and                    

.                           softening of plastics. However, the ability to degrade synthetic                   

.                           polymers by acidophilic bacteria has been scarcely investigated. …..  

v Replaced

Page 13 Lines 33 

 - 35:                                                                       ……. hyperalkaline waters (pH 11)             

.                       from the spring Poon Bate, Betelan, Zambales, Philippines [51]. Nine               

.                       strains were isolated after enrichment at a synthetic medium                                      

.                       supplemented with ldPE as a sole source of carbon pH 11 

Replace with:                                         ……. hyperalkaline water samples (pH 11)                  

.                              from a spring in the Philippines [51]. Nine strains were isolated after                

.                              enrichment in a synthetic medium supplemented with ldPE as a sole              

.                               source of carbon and adjusted to pH 11 

Replaced

Page 13 Lines 38          

- 40:                                 …… and a Bacillus sp.  Each of them separately reduced                

.         the polymer weight by up to 9.9%, 8.3%, 5.1%, and 6.3% respectively for 90           .                  

days without any pre-treatment of ldPE. A slow and constantly 

Replace with:      … and an unclassified Bacillus sp. Pure cultures of the                              

.                   isolates reduced the polymer weight by up to 9.9%, 8.3%, 5.1%, and              .         

6.3% respectively after 90 days of growth without any pre-treatment of ldPE. In          .          

each case, a slow and constantly 

Replaced

Page 13 Line 45:    the strains at pH 11. The weight of the residual polymer reduced ….. 
Replace with:     the participating strains at pH 11. The weight of the residual polymer was     

.                           reduced ….. 

Replaced

Page 13 Lines 48 

52:                                            ….. isolated from the biofilm and the effectiveness of                     

.              degradation by pure strains was investigated. A degree of 6.46%  and              

.               8.36% polymer weight loss was observed for the pure strains                         .   

                correspondingly in the presence of IONPs and 9.62% and 11.32%, 

                 correspondingly in the presence of IONPs. Comparison of the 

Replace with:  ….. isolated from the biofilm , both of which are classified as obligate.         

.       alkaliphilic Bacilliaceae. The effectiveness of degradation of PE by the pure strains    

.        was investigated. Corresponding polymer weight losses of 6.46% and 8.36%          

.        were observed for pure cultures of the isolates in the presence of IONPs , which     

.        increased to 9.62% and 11.32% in the presence of IONPs. Comparison of the 

Replaced

Page 14 Lines1 

5:        degradation degree revealed highest efficiency for unformulated bacterial              

.          .consortium, followed by formulated consortium. Lowest efficiency  was                         

.           observed for individual  strains.                                                                                                         

.                    The identified alkaliphilic plastic degraders belong to the obligate                  

.           alkaliphilic species of the family Bacilliaceae. 

Replace with:            The same albeit enhanced trend was observed with both the                     

.           unsupplemented and IONP-supplemented biofilm community, which even in the        

.           absence of the iron oxide nanoparticles was more effective than either of the        

.           strains isolated from the community.   

                           Currently available information suggests that most dentified                      .                
alkaliphilic plastic degraders are obligate alkaliphilic species of the family                   .                

Bacilliaceae, a trend shared with the known thermophilic plastic degrading           .                

eubacteria. 

Replaced

Page 14 Line 7                                                                                                                         

- 9:                          There is a lot of hope in the bioremediation of marine environments         

.                               and salt niches like plastic polluted industrial wastewaters and salt         

.                               marshes. 

Replace with:        There is growing  evidence for the bioremediation of plastics in marine.  

.                          and other natural saline environments such as salt marshes, as well as in        

.                             salt-rich industrial wastewaters. Most of the characterised competent    .                           

microorganisms have been found to be moderate or only slight               .                              

extremophiles, with species of the genus Erythrobacter being                  .                              

predominant. Some examples of halophilic marine microorganisms            .                              

able to degrade plastics are shown in Table 2. Significantly, the role of     .                            

multi-species microbial biofilms in promoting plastic degradation in such                       .                            

environments has become increasingly recognised in recent years. 

Replaced

Page 14 Line 10 

15:                     The presence of different microbial …………. 

                                ………………PE (72.2%), followed by PP (18.0%), and PS 

Replace with:         A survey of different niches of seawater in the Western Mediterranean     

.                        Sea [  ] consistently found that in each sampled area not only the highest      

.                        number but also the highest density of bacteria were detected attached to 

…                      plastic debris when compared with both sessile bacteria attached to          .                         

other organic particles and free living bacteria. The plastic debris typically    .                         

consisted of PE (72.2%), followed by PP (18.0%), and PS 

Replaced

Page 14 Line16:                  ……. FTIR analysis. Dominant 

Replace with:                        ……. FTIR analysis. By characterising the operational               

.                            taxonomic units (OTUs) in the total DNA extracted from each of the        

.                             samples, it was found that the dominant  

Replaced

Page 14 Lines 20 

25:                11.1%, respectively), while OTUs from organic particle …………. 

                                  ………………………….    Large presence of Cyanobacteria 

Replace with:          11.1% respectively) .  In contrast, equivalent analyses confirmed that 
the dominant microorganisms sourced from plastic debris were Cyanobacteria (40.8%, 
mainly Pleurocapsa sp.) and Alphaproteobacteria (32.2%, mainly Roseobacter sp. and 
Erythrobacter sp.), while the dominant microorganisms sourced from other organic 

particles were Alphaproteobacteria (25.9%, mainly Erythrobacter sp.), 
Gammaproteobacteria (25.0%, mainly Alteromonas sp.) and Cyanobacteria (17.9%, 

mainly Synechococcus sp.). It was suggested that the relatively large recorded presence 

of Cyanobacteria species on plastic debris was not only determined by their established 
important role in biofilm formation [ * ], but also possibly by some activity towards plastic 
debris: specifically identified strains belonged predominantly to two genera, Calothrix sp. 
and Pleurocapsa which are known halophiles often isolated from marine environments. 
Analysis of the OTUs sourced from plastic debris also confirmed that Erythrobacter 

species were predominant (43%) amongst the detected hydrocarbonoclastic bacteria, and  that two other moderate halophilic genera, Hyphomonas and Phorimidium, were present in  significantly higher levels than in samples sourced from the other two environments.  

Replaced

                   [ * ].  Additional new reference required to support this statement 

Reference Rossi, F., & De Philippis, R., 2015 was included

Page 14 Line 36:        Comparison on the …….  Replace with:              Comparison of the ….. 

Replaced

Page 14 Line39:    composition {54]. The established salinity suggests active growth of 
Replace with:        composition [54]. It was suggested that the significant difference in           

.                             salinity may be reflected in the active growth of 

Replaced

Page 14 Lines 42 

43:                         the V3-V4 region of the 16S rRNA gene study of the microbial                 

.                            composition revealed different presented genera. Erythrobacter 

Replace with:        the V3-V4 region of the 16S rRNA genes was used to characterise     

.                             the microbial composition of the biofilms. Erythrobacter

Replaced

Page 14 Line 49 

50:                         removal of biofilms during the traced time intervals resulted in a            

.                       more intense biodegradation of the polymer. The dominant genera in PS 

Replace with:       removal of biofilms throughout the 60-day incubation period resulted      .                            in a more intense biodegradation of the polymer. In comparison, the      ,                            dominant genera in PS 

Replaced

Page 15 Line 4 

-10:                   the basis of the iVikodak package analysis. Based on the metabolic              

.                       pathways of different taxa this enzyme was affiliated with the genera.          

.                       Pseudomonas, Arenimonas, and Acidovorax in industrial water samples          

.                       and Erythrobacter, Maribacter, and Mycobacterium in sea water samples       

.                       As the same enzyme is involved in phenylalanine metabolism and as the          

.                       mechanism of PS degradation is poorly understood, this hypothesis                

.                       should be further confirmed. 

Replace with:        the basis of iVikodak metagenomics analysis [ * ]. Based on the                

.                             known metabolic pathways of different taxa , this enzyme was most       

.                            likely to be affiliated with detected species of the genera Pseudomonas,   

.,                           Arenimonas, and Acidovorax in the industrial water samples  ,  and           

.                            with detected species of the genera Erythrobacter, Maribacter, and         

.                            Mycobacterium in the sea water samples. As the same enzyme  is          

.                             known to be involved in phenylalanine metabolism [ ** ],  but its               

.                             relationship to the mechanism of PS degradation is poorly understood,    

.                            this requires further clarification. 

Replaced

                              [ * ]  Additional new reference required for this methodology   

A reference Nagpal et al., 2019 was included

                            [ ** ]  Additional new reference required

A reference Schmitt et al., 2017 was included

Page 15 Lines12 

13:                                         …… industrial waters with salinity 18.6, 0.09-0.3, and 1.3                             

.                           g/L , correspondingly [55]. Investigation of microbial diversity in these  

.                           consortia revealed a universal …….. 

Replace with:     …… industrial waters with respective salinity of 18.6, 0.09-0.3, and 1.3       

.               g/L [55]. Investigation of microbial diversity in consortia isolated from these       

.                           environments revealed a universal …….. 

Replaced

Page 15 Line 18:   and Acidovorax due to the presence of genes responsible …. 

Replace with:        and Acidovorax due to the detected presence of genes responsible …. 

Replaced

Page 15 Line 23:                 Biofilm formation on PCL and PVA ………… 

Replace with:                      The involvement of sessile microbial communities in the                       

.                              biodegradation of other types of plastic has been                                    

.                              recognised.  Biofilm formation on PCL and PVA ………… 

Replaced

Page 16 Lines 5 

16:                              Approximately71% of the Earth is covered …… 

.                            ……………………… microbal community composition and  function       .                        

[25]. Attachment to surface and growth within biofilms is an important 

Replace with:          Approximately 70% of the Earth is covered by marine water, 90% of     

.             which maintains a constant temperature of ~5oC regardless of latitude [57]. As a   

.              result, a predominant part of the Earth’s surface provides a suitable environment    

.             for the development of psychrophiles. Often marine microorganisms are multiple   

.             extremophiles, such as psychro-halophiles, psychro-piezophiles, or psychro-       .               

alkaliphiles [57]. Most of the plastic degrading bacteria identified in cold               .               

environments belong to the genera Shewanella, Moritella, and Psychrobacter.      .               

Some examples of psychrophilic marine microorganisms able to degrade             .               

plastics are shown in Table 2. While it has been proposed that                                .               

such microorganisms play a role in degrading the increasing presence of                .              

plastics in marine ecosystems there is currently insufficient knowledge to              .              

understand the influence of plastics on microbial life, function, and community       .             

structure in these environments [25]. However, it is recognised that attachment to    .              

surfaces and growth within the resultant biofilms is an important 

Replaced

Page 17 Line 7 

16                                         ……………….(no more than 16oC) that suggests a                                     

                                                     ……………………PU-A after 12 months incubation. 
Replace with:            ……………….(no more than 16oC) , and is thus favourable for the 

.                             growth of psychrophilic bacteria. Consequently, it was chosen as  the    

.                             environment to compare the biodegradability of uncrosslinked                  

.                              poly(ethylene-butylene-adipate) (PU-A) and slightly crosslinked poly-    

‘                              (ε-caprolactone) (PU-B), two poly(ester-urethane) plastics with                 

.                             different  structural characteristics [19]. Samples of both plastics were    

.                            exposed to the seawater in Gdansk harbour (about 10oC and pH ~8). After 12 months a loss of tensile strength, discolouration, and cracking    

.                            were recorded with both materials, but more significant for                          

.                             uncrosslinked PU-A. 

Replaced

Page 17 Line 23:     ….. is performed with higher rate compared to those in the  Replace with:        ….. is performed at a higher rate than in the

Replaced

Page 17 Line 29:    Investigation on temperature and hydrostatic pressure effects on the  Replace with:   Investigation of the effects of temperature and hydrostatic pressure on the

Replaced

Page 17 Line 30:      …………..revealed that Shewanella and Moritella ….. 
Replace with:                            revealed that all the  Shewanella and Moritella ….. 

Replaced

Page 18 Table 2.  This Table needs to be reformatted. Where it is not possible to include     

.                             whole words, they need to be hyphenated in a ‘sensible’ way, eg 
                              Degrad-       Biodegrad-  or  Bio-                    Refer-   or   Ref- 
                               ation             able.                degradable         ence           erence

Reformatted 

Page 19 Line 4 

5:                                                       ………………….. and growth profiles in   
                              hydrostatic pressures an active ……… 

Replace with:                                  ………………….. and growth profiles under different                                 hydrostatic pressures  ,    an active ……… 

Replaced

Page 19 Line 8;   the cited reference for this research should be Oliveira et al 2021, not                

.                            Oberbeckmann et al 2018

Corrected with Sekiguchi et al., 2011.

Page 19 Line 8:      reliable way …….  Replace with:         promising way …… 

Replaced

Page 19  Line 21 

22:                                                           ……Despite the short evolution time, 
                                    they have adapted ………………………. 

Replace with:                                     ……Despite the relatively short evolution time, 

                         a significant number of extremophilic microorganisms have adapted …. 

Replaced

Page 19.Line 26:     …..their enzymes (the rigidity of the molecule limits the …. 

Replace with:     ………their enzymes (resulting from the rigidity of some plastic                                   

.                                                             polymers which limits the …… 

The phrase was clarified

Page 19 Line 30:     this , i , the very extreme …….. 
Replace with:           this : firstly, the very extreme ….. 

Replaced

Page 19: Line31:                 ……………………….. degrade plastic; ii, the 

Replace with:                        ………………………..degrade plastic , and secondly the 

Replaced

Page 19 33:     not be supported by such hard degradable polymers. Probably, after a   
Replace with:   not be totally supported by such hard to degrade polymers. It is possible      

.                         that, after a …… 

Replaced

Page 19 Lines 35 

36:                                                                                  …………….is quite                          

.                      promising due to their ……………….. 

Replace with:                                                                  ……………..  is a 

                       promising option due to their ……… 

This phrase was removed

Page 19 Line37:     …. be developing of metagenome technology that permits …. 
Replace with:         …. be developing relevant metagenome technologies that permit …. 

Replaced

Page 19 Line 38 

39:                                … novel enzymes. or such with improved properties directly in                           

.                              environmental metagenomes, as well as ……… 

Replace with:         ……..novel enzymes or variants of known enzymes with improved              

.                              relevant properties, as well as …….. 

Replaced

Page 19 Line41:      bacteria and their properties determined the object of the current ….. 

Replace with:           bacteria and their properties was the objective of the current …….. 

Replaced

Reviewer 2 Report

In this work, the authors present a review on plastic degradation by extremophilic bacteria. This is a very hot topic because plastic is one of the most significant pollution problems of our time.

First of all, they should have entered the number of lines to make the reviewers' job easier.

  1. General features of plastic degradation

I detect thought the text many formatting problems that the authors have to fix

In the Enzymes participating in plastic degradation part, the authors should provide the enzymes capable of degrading plastics with a more precise organization, also with the aid of a table, and also give some more precise examples of enzymes (derived from bacteria) capable of attacking plastics

paragraph 1.4 confuses me a bit. in the previous paragraph we talked about enzymatic activities, in this one instead of a "standard" method for the detection of biological activities against plastic. I believe the authors should reconsider this paragraph

  1. Extreme environments and extremophiles

In the first part of this paragraph, the authors should provide broader bibliographic support, providing examples of the most abundant microorganisms in extreme environments and provide some more indication of which kingdoms are found in these harsh environments.

Minor issues:

Fig 2 caption: close the ) after the reference.

  1. Alkaliphilic degraders, I think this point is 2.1 or 4, but in this case, you have to fix the numeration

Fig 5 всневщ ???? and provide the reference of this figure

Author Response

In this work, the authors present a review on plastic degradation by extremophilic bacteria. This is a very hot topic because plastic is one of the most significant pollution problems of our time.

First of all, they should have entered the number of lines to make the reviewers' job easier.

  1. General features of plastic degradation

I detect thought the text many formatting problems that the authors have to fix

I have checked carefully for formatting problems and hope will not appear again. 

In the Enzymes participating in plastic degradation part, the authors should provide the enzymes capable of degrading plastics with a more precise organization, also with the aid of a table, and also give some more precise examples of enzymes (derived from bacteria) capable of attacking plastics

According to the reviewers recommendation a table (Table 1) was included. Enzymes were arranged with a more precise organization with an indication of the enzymes classes and bacterial sources.

paragraph 1.4 confuses me a bit. in the previous paragraph we talked about enzymatic activities, in this one instead of a "standard" method for the detection of biological activities against plastic. I believe the authors should reconsider this paragraph

Paragraph 1.4 place was changed (1.3) and enzyme assays discussed in the light of one of the standard methods for evaluation of biodegradation in the next paragraph. 

  1. Extreme environments and extremophiles

In the first part of this paragraph, the authors should provide broader bibliographic support, providing examples of the most abundant microorganisms in extreme environments and provide some more indication of which kingdoms are found in these harsh environments.

Examples of the most abundant genera from the kingdom Bacteria were provided for each type of extreme environments supported by bibliographic information.

Minor issues:

Fig 2 caption: close the ) after the reference.

Done

  1. Alkaliphilic degraders, I think this point is 2.1 or 4, but in this case, you have to fix the numeration

The numeration was corrected

Fig 5 всневщ ???? and provide the reference of this figure

Slavic letters were removed

Reviewer 3 Report

I have no major criticism to this paper. It looks like a nice review. Given the glass transition temperature of many plastic polymers, thermophilic enzymes have emerged as promising biocatalysists for polymer degradation. The authors bring in other extreme environmental factors that may also affect the polymer structure (pH, salinity, etc.) 

I can just point at some suggested changes/amendments.

  • The authors briefly refer to acidophiles and piezophiles in this review and the lack of reports for plastic degradation in these environments and therefore, they are not covered. Given that both pressure and pH (and oxidative stress in some extreme acidic environments) may also contribute to the polymer estability, do they authors foresee any relevant findings coming from these type of environments in the near future?
  • Maybe the authors want to make a reference to microplastics when they talk about biodeterioration.
  • When talking about groups of enzymes, I would probably firstly refer to enzyme classes (hydrolases and oxidoreductases) and then mention the groups. This is because both esterases, proteases and cutinases are all hydrolases. Moreover, cutinase is a special type of esterase.
  • Eukaryote is probably the right form (not eucaryote).
  • Maybe you want to refer to "facultative thermophiles" as "thermotolerant"?
  • Other relevant thermophilic environments include the deep terrestrial subsurface, or even dry arid soils can host thermophiles. As for low temperature environments, I would probably mention sea ice or deep sea waters as very relevant for this subject.
  • "subsp" is normally not in italics (check both Table 1 and main text).
  • If you include the phylum/class of all others, then both Arenimonas and Pseudomonas belong to Gammaprotebacteria.
  • Phenylacetaldehyde dehydrogenase is EC 1.2.1.39
  • Black Sea, both words are capitalized.
  • Consider to use the full name the first time you name UNEP.
  • Table 2. The column "effectiveness of degradation" should be completed. If it is not reported, consider to write "not reported".
  • The authors should consider citing the work by Sulaiman et al., 2012, where the report an polyester hydrolase from a metagenome obtained from compost, which is also slightly thermoestable (10.1128/AEM.06725-11).

Author Response

I have no major criticism to this paper. It looks like a nice review. Given the glass transition temperature of many plastic polymers, thermophilic enzymes have emerged as promising biocatalysists for polymer degradation. The authors bring in other extreme environmental factors that may also affect the polymer structure (pH, salinity, etc.) 

I can just point at some suggested changes/amendments.

  • The authors briefly refer to acidophiles and piezophiles in this review and the lack of reports for plastic degradation in these environments and therefore, they are not covered. Given that both pressure and pH (and oxidative stress in some extreme acidic environments) may also contribute to the polymer estability, do they authors foresee any relevant findings coming from these type of environments in the near future?

The next sentences were included:

“…are not available although the decreased plastic strength in acidic environment suggests a good prospect for faster waste degradation. In the same time acidic pH shorten life of some plastic products used in bleaching processes.

and

In the same time a decrease in plastic strength by increased pressure at depth water should be consider.”

  • Maybe the authors want to make a reference to microplastics when they talk about biodeterioration.

Microplastics with their peculiarities in the mechanism of degradation are not an object of the current review.

  • When talking about groups of enzymes, I would probably firstly refer to enzyme classes (hydrolases and oxidoreductases) and then mention the groups. This is because both esterases, proteases and cutinases are all hydrolases. Moreover, cutinase is a special type of esterase.

Enzymes were arranged with an indication of the enzymes classes and then their groups in Table 1.

  • Eukaryote is probably the right form (not eucaryote).

Done

  • Maybe you want to refer to "facultative thermophiles" as "thermotolerant"?

The next sentences were included:

“Facultative thermophiles thrive at temperatures 41-50 °C, while thermotolerant microorganisms are mesophilic microorganisms that can tolerate temperature higher than 41°C however grow optimally at lower temperature.”

  • Other relevant thermophilic environments include the deep terrestrial subsurface, or even dry arid soils can host thermophiles. As for low temperature environments, I would probably mention sea ice or deep sea waters as very relevant for this subject.

As an information concerning plastic degradation in the deep terrestrial subsurface, dry arid soils and sea ice was not found, they were not included in the review. The next phrase was included in the sentence:

Cold environments comprise fresh and marine waters including deep sea water,…”

  • "subsp" is normally not in italics (check both Table 1 and main text).

Corrected

  • If you include the phylum/class of all others, then both Arenimonasand Pseudomonas belong to Gammaprotebacteria.

Included

  • Phenylacetaldehyde dehydrogenase is EC 1.2.1.39

Added

  • Black Sea, both words are capitalized.

Corrected

  • Consider to use the full name the first time you name UNEP.

Full name “United Nations Environment Programme” was used in the first mention.

  • Table 2. The column "effectiveness of degradation" should be completed. If it is not reported, consider to write "not reported".

Table 2 was completed.

  • The authors should consider citing the work by Sulaiman et al., 2012, where the report an polyester hydrolase from a metagenome obtained from compost, which is also slightly thermoestable (10.1128/AEM.06725-11).

Cutinase work of Sulaiman et al., 2012 was included.

Reviewer 4 Report

I really liked this well-written and comprehensive review on plastic degradation by extremophilic bacteria. This is a critical subject for environmental studies now days. Unfortunately, the data on this matter is scarce, the processes are understudied and very far from real understanding. Yet, the input and impact of plastics in the environment are enormous.

The Authors did a great job collecting all available data on plastics degradation in extreme environments, such as high and low temperatures, high alkalinities and high salt. It could be very helpful and might promote research in this field.

I have some minor comments mostly regarding English language.

Paragraph 1.2 “relative deal of crystalline and amorphous…” Use “share” instead of deal

                            “size and shape of substrates” use “form” instead of shape

                             “plastics are divided into two main groups” use colon (:) instead of comer.

                              Use comer before “which”

                           “are further degraded by secondary degraders into their cells” Remove “into             their cells”

“oligomers” after dash – small letter.

“approach for resolving of plastic disposal problems” remove “of”

“microorganisms; gene expression…” Use “or” instead of semicolon

Paragraph 1.3

“the resistance of plastics against microbial attack” Use resistance to microbial attack

“ but to the attack to the chemical additives in their molecules” Is it “attack OF chemical additives ON their molecules”? Not clear sentence.

Paragraph 1.4

“5. Registration of products” It is better to use “detection” instead of “registration”.

“showed different success” It’s better to use “variable” instead of “different”.

“formed biofilms which was related with degradation of PE, biofilms by….”

Use “…that were related to degradation of PE.” A dot here. Then start a new sentence

and use either “biofilms formed BY” or “biofilms of Ps. Xxxx”. Start again a new sentence on B. subtilis and B. cereus.

Part 2.

“Growth of hyperthermophiles is best at above 80C [31]. Is it right reference? Check ,please!

The same about reference [32].

“osmotic pressure balance reveived byorganic compatable solutes” Did you mean “achieved”?

Again, check the ref. [32].

Table 1 and Table 2. First column should be called “Plastic degradation type”. Degradation type refers to a chemical reaction.

“Sequence differences” Did you mean aminoacids sequence differences? Should be clarified.

“two the best of our knowledge” is used two times. Remove, please.

Wrong numbering of the chapters. “Alkaliphilic degraders “is numbered 1. Yet, the previous chapter on Extremophiles was numbered 2. Make the numbering uniformed throughout the paper, please!

“The addition of IONPs facilitated biofilm formation.” Is it any explanation to this?

4.Halophilic degraders.

“Two other moderate halophilic genera…… were 10-100 more found….”

Use “Two other moderately hallophiliuc genera were 10-100 more abundant”

“ in comparison with other two communities” Its better to clarify what communities than to send a reader to go looking for that in the paper.

“iVikodak package analysis” requires a link in brackets.

Like other extremophiles halophilic producers of plastic degrading enzymes are slight and moderate extremophiles” requires a comer after “other extremophiles”

The above statement sounds like a conclusion. In this respect, next sentence looks out of place. Not logical.

Conclusions

Use “human” instead of “man  life”.

Author Response

I really liked this well-written and comprehensive review on plastic degradation by extremophilic bacteria. This is a critical subject for environmental studies now days. Unfortunately, the data on this matter is scarce, the processes are understudied and very far from real understanding. Yet, the input and impact of plastics in the environment are enormous.

The Authors did a great job collecting all available data on plastics degradation in extreme environments, such as high and low temperatures, high alkalinities and high salt. It could be very helpful and might promote research in this field.

I have some minor comments mostly regarding English language.

Paragraph 1.2 “relative deal of crystalline and amorphous…” Use “share” instead of deal

Done

                            “size and shape of substrates” use “form” instead of shape

Done

                             “plastics are divided into two main groups” use colon (:) instead of comer.

Done

                              Use comer before “which”

Done

                           “are further degraded by secondary degraders into their cells” Remove “into             their cells”

Removed

“oligomers” after dash – small letter.

Done

“approach for resolving of plastic disposal problems” remove “of”

Done

“microorganisms; gene expression…” Use “or” instead of semicolon

Done

Paragraph 1.3

“the resistance of plastics against microbial attack” Use resistance to microbial attack

Done

“ but to the attack to the chemical additives in their molecules” Is it “attack OF chemical additives ON their molecules”? Not clear sentence.

The phrase was changed next way:

“…to the attack of the chemical additives.”

Paragraph 1.4

“5. Registration of products” It is better to use “detection” instead of “registration”.

Changed

“showed different success” It’s better to use “variable” instead of “different”.

Changed

“formed biofilms which was related with degradation of PE, biofilms by….”

Use “…that were related to degradation of PE.” A dot here. Then start a new sentence

and use either “biofilms formed BY” or “biofilms of Ps. Xxxx”. Start again a new sentence on B. subtilis and B. cereus.

 Done

Part 2.

“Growth of hyperthermophiles is best at above 80C [31]. Is it right reference? Check ,please!

Corrected

The same about reference [32].

Corrected

“osmotic pressure balance reveived byorganic compatable solutes” Did you mean “achieved”?

Changed

Again, check the ref. [32].

Table 1 and Table 2. First column should be called “Plastic degradation type”. Degradation type refers to a chemical reaction.

Corrected

“Sequence differences” Did you mean aminoacids sequence differences? Should be clarified.

Aminoacid was added

“two the best of our knowledge” is used two times. Remove, please.

Removed

Wrong numbering of the chapters. “Alkaliphilic degraders “is numbered 1. Yet, the previous chapter on Extremophiles was numbered 2. Make the numbering uniformed throughout the paper, please!

Numbering was corrected.

“The addition of IONPs facilitated biofilm formation.” Is it any explanation to this?

The authors explanation was included:

The effect of IONPs was attributed to the properties of the nanoparticles such as magnetism and electrostatic charge leading to alter bacterial motion through signal transduction as well as the formation of cofactors produced in the medium.  As a result higher hydrophobicity of the consortium with IONPs and higher adhesion to the plastic surface was demonstrated.

4.Halophilic degraders.

“Two other moderate halophilic genera…… were 10-100 more found….”

Use “Two other moderately hallophiliuc genera were 10-100 more abundant”

Changed

“ in comparison with other two communities” Its better to clarify what communities than to send a reader to go looking for that in the paper.

Clarified

“iVikodak package analysis” requires a link in brackets.

A reference was included.

Like other extremophiles halophilic producers of plastic degrading enzymes are slight and moderate extremophiles” requires a comer after “other extremophiles”

The above statement sounds like a conclusion. In this respect, next sentence looks out of place. Not logical.

This paragraph was changed next way:

The moderate halophilic representatives of the genus Erythrobacter are often identified as an actively growing in plastic.  Like other extremophiles, halophilic producers of plastic degrading enzymes are slight and moderate extremophiles.

Conclusions

Use “human” instead of “man  life”.

Done

Round 2

Reviewer 1 Report

The following need attention:

a. Page 8:  The concluding paragraph of section 1.3.2. (Enzymes participating in ... etc) should include the addition previously suggested, ie:

It is known that extremophilic microorganisms are competent producers of a range of potentially relevant hydrolytic enzymes[Kour et al 2019].

The reference 'Kour et al 2019' will now become [42], and all current references [42] et seq will need to be renumbered accordingly as [43] et seq both in text and in the Bibliography.

The full details of this reference are:

'Extremophiles for hydrolytic enzyme production: biodiversity and potential biotechnological applications'

D.Kour, K.L.Rana, T.Kour, B.Singh, V.S.Chauhan, A.Kumar, A.A.Rastegan, N.Yadav, A.J.Nadav, and V.K.Gupta.

pp321 -372 in Bioprocessing for Biomolecules Production

Edited by G.Molina, V.K.Gupta, B.N.Singh, and N.Gathergood.

John Wiley & Sons, November 2019

doi:10.1002/9781119434436.ch16

b. Page 12, Table 2:                                                                                                    i.     Mw   replace with MW   (one example of this remains in text).                                                                                                          ii.    Polycaprolacto  ne  replace with Polycapro lactone

c. Page 19, Table 3:     Plastic degradatio n type replace with Plastic degradation type.                                                                                              Ideally, should also replace Biodegr arable with Biodegrad  able, but this will possibly require reformatting the Table

Author Response

  1. Page 8:  The concluding paragraph of section 1.3.2. (Enzymes participating in ... etc) should include the addition previously suggested, ie:

It is known that extremophilic microorganisms are competent producers of a range of potentially relevant hydrolytic enzymes[Kour et al 2019].

The reference 'Kour et al 2019' will now become [42], and all current references [42] et seq will need to be renumbered accordingly as [43] et seq both in text and in the Bibliography.

The full details of this reference are:

'Extremophiles for hydrolytic enzyme production: biodiversity and potential biotechnological applications'

D.Kour, K.L.Rana, T.Kour, B.Singh, V.S.Chauhan, A.Kumar, A.A.Rastegan, N.Yadav, A.J.Nadav, and V.K.Gupta.

pp321 -372 in Bioprocessing for Biomolecules Production

Edited by G.Molina, V.K.Gupta, B.N.Singh, and N.Gathergood.

John Wiley & Sons, November 2019

doi:10.1002/9781119434436.ch16

The reference 42 was included and the numbers of the next references were changed.

  1. Page 12, Table 2:                                                                                                    i.     Mw   replace with MW   (one example of this remains in

text).  

   Mw was replaced by MW                                                                                                

  1.    Polycaprolacto  ne  replace with Polycapro lactone

The used Page setup for Table 2 and 3 in the variant I sent was Landscape and the word Polycaprolactone is on one row. I cannot understand why the word is separate. That’s why I am sending now also pdf format.

  1. Page 19, Table 3:     Plastic degradatio n type replace with Plastic degradation type.                                                                                              Ideally, should also replace Biodegr arable with Biodegrad  able, but this will possibly require reformatting the Table

The used Page setup for Table 2 and 3 in the variant I sent was Landscape and the word Polycaprolactone is on one row. I cannot understand why the word is separate. That’s why I am sending now also pdf format.

Reviewer 2 Report

The authors adequately revised the paper. it is my opinion that it can be accepted in the current version

Author Response

The authors adequately revised the paper. it is my opinion that it can be accepted in the current version.

As the reviewer has no other comments except paper acceptance, I support his suggestion.